# Detecting the Semantic Fixed Point: A Geometric Framework for Efficient Inference

**Jiawei Gu** [1]  **Ziyue Qiao** [1]  **Xiao Luo** [2]

## Abstract

Each layer of a Transformer refines the hidden state toward a prediction, an iterative process resembling fixed-point iteration. Yet when should this iteration terminate? Existing early exit methods rely on output confidence as a proxy for internal convergence. We take a more direct approach by examining the geometry of the hidden state trajectory. We find that layer-wise updates exhibit a two-phase structure: large, volatile updates in early layers, followed by small, aligned updates as the model propagates an already-formed representation. The transition is remarkably sharp. This yields a simple criterion: exit when step size vanishes and direction stabilizes. We track the normalized update norm and cosine similarity between consecutive updates, exiting when both indicate convergence. The overhead is $O(d)$ per layer, independent of vocabulary size, requiring no learned components or architectural modifications. On LLaMA-2-7B and LLaMA-2-13B across question answering and commonsense reasoning tasks, this geometric criterion reduces FLOPs by 30–35% while retaining over 97% of full-depth accuracy.

## 1. Introduction

Transformer inference has become increasingly expensive, with large language models requiring substantial computational resources for every forward pass (Kaplan et al., 2020; Brown et al., 2020). Early exit methods (Teerapittayanon et al., 2016; Schwartz et al., 2020) offer a promising approach to reducing this cost by terminating computation when further layers yield diminishing returns. Yet as Trans-

[1] School of Computing and Information Technology, Great Bay University, Dongguan, China [2] Department of Statistics, University of Wisconsin-Madison, USA. Correspondence to: Ziyue Qiao <ziyuejoe@gmail.com>.

*Proceedings of the 43rd International Conference on Machine Learning*, Seoul, South Korea. PMLR 306, 2026. Copyright 2026 by the author(s).

formers scale to billions of parameters with vocabularies exceeding 32,000 tokens, traditional confidence-based early exit approaches reveal fundamental limitations in terms of computational efficiency that demand a rethinking of how we determine when inference computation should stop.

**The iterative nature of Transformer depth.** Each layer of a Transformer refines its hidden state, nudging the representation closer to a form suitable for prediction. Writing $\mathbf{h}^{(l)}$ for the hidden state at layer $l$, the residual structure yields

$$\mathbf{h}^{(l)} = \mathbf{h}^{(l-1)} + F_l(\mathbf{h}^{(l-1)}), \qquad (1)$$

where $F_l$ encapsulates the attention and feed-forward operations. This formulation echoes the classical fixed-point iteration $x_{k+1} = g(x_k)$ from numerical analysis, with one important distinction: the mapping $F_l$ varies across layers, making this a *non-stationary* iterative process. In numerical methods, iterative procedures have principled termination criteria: one halts when successive updates become negligible, signaling that the solution has stabilized. Yet in deep learning, we answer the termination question with a fixed constant $L$, the number of layers, applied uniformly regardless of input complexity. A simple factual lookup and a complex multi-hop reasoning problem traverse identical computational paths. This is a paradigm mismatch: *a static architectural constant governs an inherently dynamic computational process that varies per input.*

**The confidence proxy problem.** The early exit literature has long recognized this inefficiency and sought to restore input-adaptive computation. However, nearly all existing methods share a common strategy: they use *output confidence* as a proxy for internal convergence. Softmax entropy methods exit when the output distribution becomes concentrated (Xin et al., 2020; Schwartz et al., 2020). Top-1 confidence approaches halt when the maximum probability exceeds a threshold (Schuster et al., 2022). Prediction consistency methods terminate when consecutive layers agree on their outputs (Zhou et al., 2020). The implicit assumption is uniform: if the model appears certain, its internal representation must have converged. But this assumption inverts the causal relationship. Output confidence is a *consequence* of representational convergence, not a *criterion* for it. A model can be confidently wrong when its representation settles into an incorrect basin, and hesitantly

correct when the representation has converged near a decision boundary. This raises a fundamental question: *can we develop stopping criteria that directly monitor convergence rather than its downstream symptoms in the output space?*

**The vocabulary coupling burden.** Beyond the conceptual issue, confidence-based methods incur a practical cost that grows increasingly prohibitive for large language models. Computing output confidence requires projecting the hidden state through the language model head, an operation with complexity $O(d \times V)$ where $d$ is the hidden dimension and $V$ is the vocabulary size. For modern LLMs with $V > 32,000$, this projection must be performed at every candidate exit layer. The theoretical speedup from reduced layers is substantially eroded by the overhead of confidence computation itself. The criterion designed to save computation becomes a computational burden. This exposes a critical limitation: *what stopping mechanism can achieve vocabulary-independent overhead while providing reliable convergence signals derived from internal hidden states?*

**First principles: what does convergence look like?** These observations point toward a fundamentally different approach. If we genuinely view the Transformer as an iterative system, we should assess convergence using properties of the iteration itself, not properties of an auxiliary output. What does numerical analysis teach us about recognizing convergence? Two conditions are canonical: the step size should vanish, $\|x_{k+1} - x_k\| \to 0$, and the direction of successive updates should stabilize rather than oscillate. When both conditions hold, the iteration has effectively reached its fixed point. This principle suggests a natural reframing: instead of asking "Is the model confident?" we should ask "Has the representation stopped changing?" The answer lies not in output distributions but in the geometry of the hidden state trajectory within the high-dimensional space.

**The two-phase structure phenomenon.** Investigating Transformer dynamics through this geometric lens reveals a striking pattern. We find that the layer-wise updates $\mathbf{u}^{(l)} = \mathbf{h}^{(l)} - \mathbf{h}^{(l-1)}$ exhibit a characteristic two-phase structure across architectures and tasks. In early layers, updates are large in magnitude and volatile in direction; the model is actively *constructing* its representation, integrating information and resolving ambiguities. Beyond a certain depth, the behavior shifts qualitatively: updates shrink dramatically and consecutive update vectors become nearly collinear. The model transitions from building the representation to merely *propagating* it through remaining layers with minimal modification. We term these the **construction phase** and the **propagation phase**. Crucially, this transition is remarkably sharp, occurring over just two to three layers rather than gradually. This sharpness makes the phenomenon practically exploitable: there exists a well-defined *saturation layer* $l_{\text{sat}}$ beyond which further computation con-

tributes little to the final representation. Moreover, this saturation point varies systematically with input difficulty, with simple queries saturating early and complex reasoning requiring deeper processing to fully converge internally.

**From geometric insight to stopping criterion.** This phenomenon suggests a principled early exit strategy: terminate when the representation update becomes small in magnitude and stable in direction. We track two quantities at each layer: the normalized step size $\tilde{s}^{(l)} = \|\mathbf{u}^{(l)}\|/\|\mathbf{h}^{(l)}\|$ and the directional stability $c^{(l)} = \cos(\mathbf{u}^{(l)}, \mathbf{u}^{(l-1)})$. Exit occurs when both indicate convergence for $K$ consecutive layers beyond a minimum depth. The entire procedure requires only $O(d)$ operations per layer, completely independent of vocabulary size, with no learned components and no architectural modifications. This criterion can serve as a drop-in signal module for existing early exit systems to bypass vocabulary bottlenecks.

**Contributions.** This work makes four contributions spanning phenomenon, method, theory, and empirical validation. We first identify and characterize the two-phase structure of hidden state trajectories, demonstrating that the construction-to-propagation transition is sharp, consistent across architectures, and systematically tied to input difficulty (Section 2). Building on this observation, we propose Geometric Convergence Early Exit (GCEE), a vocabulary-independent stopping criterion requiring only $O(d)$ overhead per layer and no learned components (Section 3). We provide theoretical grounding by establishing that propagation-phase updates follow approximate geometric decay, yielding explicit error bounds that guide principled threshold selection (Appendix B). Finally, experiments on LLaMA-2-7B and LLaMA-2-13B across question answering and commonsense reasoning tasks show that GCEE reduces FLOPs by 30–35% while retaining over 97% of full-depth accuracy. This translates to a $1.45\times$ end-to-end speedup, significantly outperforming the $1.14\times$ speedup of entropy-based methods whose theoretical gains are **severely** eroded by vocabulary-coupled overhead (Section 4).

**Conflict of Interest Disclosure.** The authors declare no financial conflicts of interest related to this work.

## 2. Phenomenon: Two-Phase Structure of Hidden State Trajectories

Before developing our early exit criterion, we establish the empirical phenomenon that motivates it. We examine hidden state trajectories in LLaMA-2-7B (32 layers) (Touvron et al., 2023) and LLaMA-2-13B (40 layers) using a 2,000-sample analysis subset drawn from TriviaQA (Joshi et al., 2017), Natural Questions (Kwiatkowski et al., 2019), HellaSwag (Zellers et al., 2019), and Wino-Grande (Sakaguchi et al., 2021). For each input, we

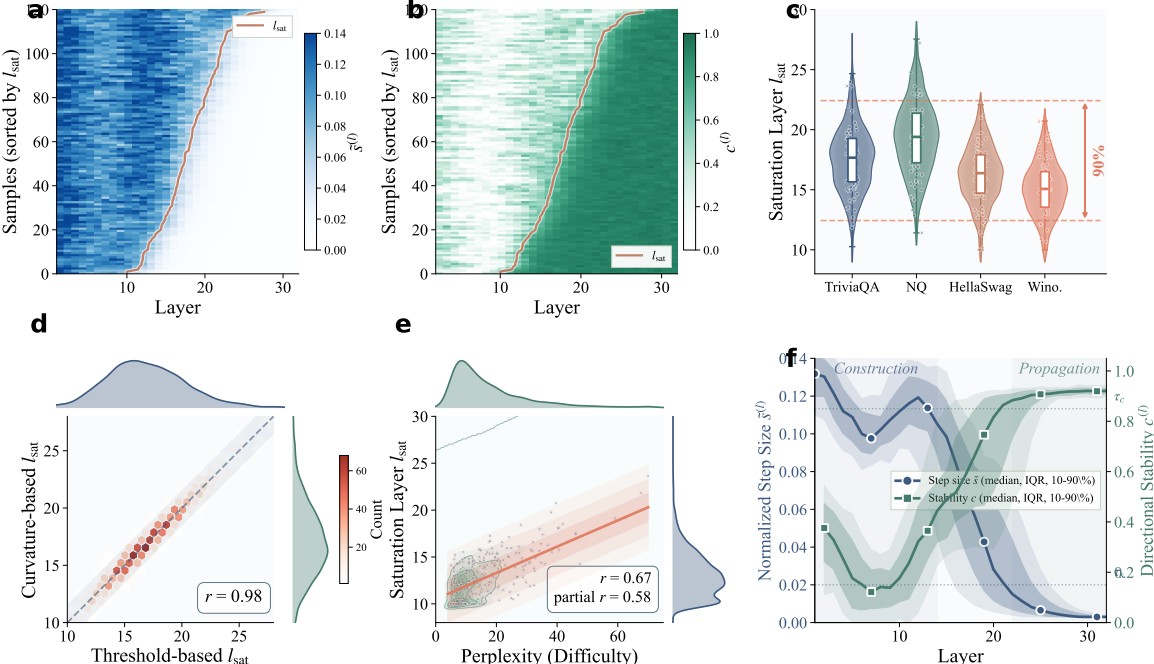

*Figure 1.* **Two-phase structure of hidden state trajectories in LLaMA-2-7B.** (a) Normalized step size $\tilde{s}^{(l)}$ for 120 samples sorted by the curvature-based saturation layer $l_{\text{sat}}$. (b) Directional stability $c^{(l)}$ for the same samples. (c) Distribution of curvature-based $l_{\text{sat}}$ across datasets. (d) Agreement between the threshold-based proxy and curvature-based detector ($r = 0.98$). (e) Correlation between input difficulty and $l_{\text{sat}}$ ($r = 0.67$). (f) Aggregate median and percentile statistics.

track the last token's hidden state $\mathbf{h}^{(l)}$ at every layer and compute two geometric quantities: the *normalized step size* $\tilde{s}^{(l)} = \|\mathbf{u}^{(l)}\|/\|\mathbf{h}^{(l)}\|$ and the *directional stability* $c^{(l)} = \cos(\mathbf{u}^{(l)}, \mathbf{u}^{(l-1)})$, where $\mathbf{u}^{(l)} = \mathbf{h}^{(l)} - \mathbf{h}^{(l-1)}$ is the update vector driving the evolution of the representation.

**Two-phase structure.** Figure 1(a,b) visualizes both quantities across 120 samples sorted by saturation layer. A consistent pattern emerges: each sample transitions sharply from high step size and low stability (construction phase) to low step size and high stability (propagation phase). The orange curve tracking $l_{\text{sat}}$ cuts cleanly through this transition. In early layers, step sizes range from 0.05 to 0.15; beyond the transition, they fall below 0.02 while stability exceeds 0.85. Figure 1(f) confirms this pattern in aggregate: variance shrinks dramatically in the stable propagation phase.

**Sharp transition.** We define the *saturation layer* $l_{\text{sat}}$ as the curvature-based transition point, computed as the layer with the maximum slope change in the smoothed $\log \tilde{s}^{(l)}$ trajectory. This definition captures the construction-to-propagation boundary independently of the stopping thresholds used by GCEE. For interpretability, we also consider a simple threshold-based proxy, defined as the first layer satisfying $\tilde{s}^{(l)} < 0.02$ and $c^{(l)} > 0.85$. Figure 1(c) shows that 90% of samples saturate within layers 14–24, with transition windows spanning only 2–3 layers. Figure 1(d)

*Table 1.* Saturation-difficulty correlation within length buckets (LLaMA-2-7B, TriviaQA full test set, $N = 11{,}313$).

| Sequence Length | Samples | Correlation |
|---|---|---|
| 64–128 | 2,845 | 0.62 |
| 128–256 | 4,521 | 0.65 |
| 256–512 | 3,102 | 0.68 |
| 512+ | 845 | 0.71 |

shows high agreement between the threshold-based proxy and the curvature-based detector ($r = 0.98$), confirming a genuine transition point rather than a threshold artifact.

**Input-dependent saturation.** The saturation layer varies with input difficulty. Figure 1(e) plots $l_{\text{sat}}$ against perplexity, revealing clear positive correlation ($r = 0.67$). This persists after controlling for sequence length (partial $r = 0.58$), confirming it reflects genuine difficulty rather than length effects. Table 1 shows the correlation holds within each length bucket, verifying the robustness of this finding.

**Summary.** Hidden state trajectories exhibit a two-phase structure: a *construction phase* of large, volatile updates followed by a *propagation phase* of small, aligned updates. The transition is sharp, input-dependent, and robustly detectable. This motivates a natural criterion: *exit when updates become small in magnitude and directionally stable.*

# 3. Method

The two-phase phenomenon identified in Section 2 suggests a principled stopping rule: terminate when representation updates become small and directionally stable. This section formalizes this geometric intuition into a concrete algorithm.

## 3.1. Refinement as a non-stationary iteration

We view each Transformer block as one step of an iterative refinement process. Let $\mathbf{h}^{(l)} \in \mathbb{R}^d$ denote the *last-token* hidden state after block $l$ ($l = 0$ corresponds to the embedding output). The residual structure yields the following recurrence

$$\mathbf{h}^{(l)} = \mathbf{h}^{(l-1)} + F_l\big(\mathbf{h}^{(l-1)}\big), \qquad l = 1, \dots, L, \quad (2)$$

where $F_l(\cdot)$ encapsulates the attention and feed-forward computations of block $l$. Unlike classical fixed-point iteration with a single mapping $g$, here each $F_l$ differs across layers, resulting in a *non-stationary* iteration. Nevertheless, the core insight from numerical analysis transfers: iteration converges when successive updates become negligible. Our goal is to find the smallest depth $l_{\mathrm{exit}}$ beyond which further refinement has negligible effect on the final prediction.

**Why the last token?** In autoregressive generation, the last token's hidden state directly determines the next-token distribution at each generation step. We therefore monitor convergence exclusively on this position. This choice is both principled and practical: it targets the causally relevant representation while requiring only $O(d)$ storage per layer. Ablations in Section 4.4 confirm that last-token monitoring outperforms alternatives such as mean pooling.

## 3.2. A vocabulary-free geometric certificate

Classical convergence criteria examine two properties: the step size should vanish, and the direction should stabilize. Rather than using confidence proxies that require evaluating logits at intermediate depths, we monitor the refinement trajectory itself using $\mathcal{O}(d)$ computations per layer on a single vector instead of the full vocabulary distribution.

**Update vector.** The layer-wise change in representation is defined as the difference between consecutive hidden states

$$\mathbf{u}^{(l)} := \mathbf{h}^{(l)} - \mathbf{h}^{(l-1)}. \quad (3)$$

This is the refinement "step" taken by the $l$-th block. In the construction phase, $\mathbf{u}^{(l)}$ is large and volatile; in the propagation phase, it shrinks and aligns with previous updates.

**Normalized step size.** Raw step magnitudes $\|\mathbf{u}^{(l)}\|$ vary substantially across inputs due to differences in hidden state scale, which can range from 2 to 3×. To obtain a threshold that transfers across prompts, we normalize by the current state norm to decouple convergence from signal magnitude:

$$\tilde{s}^{(l)} := \frac{\|\mathbf{u}^{(l)}\|_2}{\|\mathbf{h}^{(l)}\|_2 + \varepsilon_s}, \quad (4)$$

where $\varepsilon_s > 0$ is a small constant for numerical stability. Empirically, $\|\mathbf{h}^{(l)}\|$ stabilizes in the propagation phase (coefficient of variation $<0.1$; see Section 2), so $\tilde{s}^{(l)}$ remains proportional to the raw step size where it matters, preserving the validity of our theoretical error bounds (Section B).

**Directional stability.** A small step is insufficient for convergence if the direction oscillates. We measure directional consistency via the cosine similarity between consecutive updates to distinguish oscillation from true convergence:

$$c^{(l)} := \frac{\langle \mathbf{u}^{(l)}, \mathbf{u}^{(l-1)} \rangle}{\|\mathbf{u}^{(l)}\|_2 \|\mathbf{u}^{(l-1)}\|_2 + \varepsilon_s}. \quad (5)$$

When $\|\mathbf{u}^{(l)}\|_2 < \delta$ (default $\delta = 10^{-6}$), the step is negligible and direction becomes numerically meaningless; we set $c^{(l)} := 1$ in this case to avoid division by zero.

## 3.3. Geometric Convergence Early Exit

**Core criterion.** We exit when refinement becomes simultaneously (i) small in magnitude and (ii) stable in direction. Define the convergence predicate as the conjunction:

$$\mathsf{G}^{(l)} := \left[\tilde{s}^{(l)} \le \tau_s\right] \wedge \left[c^{(l)} \ge \tau_c\right]. \quad (6)$$

The conjunction is essential: step size alone permits "small oscillations," while directional stability alone permits "correct heading but far from destination." Together they provide strong evidence that the iteration has effectively and robustly stabilized (see Section B for formal justification).

**Robustness safeguards.** Two mechanisms prevent premature exits. First, a *warmup depth* $l_{\min}$ ensures the model has processed sufficient layers before exit is considered. Second, we require $K$ *consecutive* layers to satisfy $\mathsf{G}^{(l)}$, filtering transient fluctuations. The complete stopping rule is defined as the minimum layer satisfying these constraints:

$$l_{\mathrm{exit}} := \min\left\{l \in \{1, \dots, L\} : l \ge l_{\min} \wedge \bigwedge_{i=0}^{K-1} \mathsf{G}^{(l-i)}\right\}, \quad (7)$$

with $l_{\mathrm{exit}} = L$ if the set is empty. Default hyperparameters are $\tau_s = 0.01$, $\tau_c = 0.9$, $l_{\min} = 4$, and $K = 2$; sensitivity analysis in Section 4.4 shows stable performance across a reasonable range of these configurable decision thresholds.

**Output generation.** At $l_{\mathrm{exit}}$ we compute logits exactly once using the original final normalization and LM head, requiring no auxiliary classifiers or learned exit heads:

$$\mathbf{z} = \mathrm{LMHead}\big(\mathrm{LN}(\mathbf{h}^{(l_{\mathrm{exit}})})\big). \quad (8)$$

*Table 2.* **Main results on LLaMA-2-7B.** GCEE reduces FLOPs by about 34% while retaining approximately 98% of full-depth performance on average, with per-task retention ranging from about 97% to 99%. Best early-exit results in **bold**.

| Method | TriviaQA | NQ | HellaSwag | WinoGrande | Avg Layer | FLOPs↓ |
|---|---|---|---|---|---|---|
| Full Model (32L) | 64.5 | 23.8 | 76.2 | 69.5 | 32.0 | 0% |
| Random ($\mu$=16) | 41.2 | 13.8 | 57.8 | 53.9 | 16.0 | 50% |
| Unified Skip | 56.8 | 19.2 | 67.5 | 63.1 | 22.4 | 30% |
| Entropy | 58.9 | 20.4 | 70.8 | 65.5 | 21.8 | 32% |
| Top-1 Conf | 58.2 | 19.8 | 70.1 | 64.8 | 21.5 | 33% |
| Hidden Sim | 56.5 | 19.1 | 68.5 | 63.8 | 20.9 | 35% |
| **GCEE (Ours)** | **63.5** | **23.1** | **74.8** | **68.5** | **21.2** | **34%** |
| *Retention* | *98.4%* | *97.1%* | *98.2%* | *98.6%* | | |

*Table 3.* **Main results on LLaMA-2-13B.** The pattern holds at larger scale.

| Method | TriviaQA | NQ | HellaSwag | WinoGrande | Avg Layer | FLOPs↓ |
|---|---|---|---|---|---|---|
| Full Model (40L) | 68.2 | 27.5 | 79.1 | 72.4 | 40.0 | 0% |
| Random ($\mu$=20) | 44.5 | 16.2 | 60.8 | 56.8 | 20.0 | 50% |
| Unified Skip | 60.5 | 22.1 | 71.2 | 66.8 | 28.0 | 30% |
| Entropy | 63.2 | 23.5 | 73.8 | 68.9 | 27.2 | 32% |
| Top-1 Conf | 62.5 | 22.8 | 73.1 | 68.2 | 26.8 | 33% |
| Hidden Sim | 61.2 | 21.8 | 71.5 | 67.1 | 26.2 | 35% |
| **GCEE (Ours)** | **67.2** | **26.8** | **77.8** | **71.5** | **26.5** | **34%** |
| *Retention* | *98.5%* | *97.5%* | *98.4%* | *98.8%* | | |

### 3.4. Inference pipeline

**Prompt-level early exit.** We adopt a per-request strategy: during the prefill pass, $\tilde{s}^{(l)}$ and $c^{(l)}$ are computed online, and forward propagation terminates at $l_{\mathrm{exit}}$. The subsequent decoding phase uses a *truncated* model consisting of the first $l_{\mathrm{exit}}$ blocks, with a standard KV-cache layout. This design ensures structural consistency by preventing missing cache entries, while facilitating seamless integration with existing inference frameworks without requiring custom kernels. Implementation details for single-pass prefill with caching and batching strategies are provided in Section C.2.

**Computational overhead.** A key advantage of GCEE is its *vocabulary-independent* overhead. At each layer, we compute one vector difference, two norms, and one inner product. These are all $O(d)$ operations on a single $d$-dimensional vector, bypassing the expensive language model head. Table 4 compares this with confidence-based alternatives.

| Method | Per-layer cost | Vocab-coupled? |
|---|---|---|
| Entropy / Top-1 | $O(d \times V)$ | Yes |
| Hidden State Sim. | $O(d)$ | No |
| **GCEE (Ours)** | $O(d)$ | **No** |

*Table 4.* Per-layer decision overhead. For LLaMA-2-7B, confidence-based methods incur roughly $\sim 25\times$ higher measured decision latency in our implementation, while their raw projection cost scales with vocabulary size.

*Table 5.* **Exit layer accuracy** (LLaMA-2-7B, TriviaQA).

| Metric | Value |
|---|---|
| Mean $|l_{\mathrm{exit}} - l_{\mathrm{sat}}|$ | 1.4 layers |
| Median $|l_{\mathrm{exit}} - l_{\mathrm{sat}}|$ | 1.0 layers |
| $l_{\mathrm{exit}} \geq l_{\mathrm{sat}}$ | 92.8% |

This difference is practically significant: as shown in Section 4.5, entropy-based methods achieve only $1.14\times$ end-to-end speedup despite similar layer savings, because their $O(dV)$ overhead erodes the theoretical gains. GCEE achieves $1.45\times$ speedup, approaching the maximum from reduced depth by eliminating projection overhead.

**Theory pointer.** Section B provides formal analysis including: (i) error bounds on $\|\mathbf{h}^{(L)} - \mathbf{h}^{(l_{\mathrm{exit}})}\|$ under geometric decay, (ii) logit perturbation bounds through LayerNorm (Ba et al., 2016) and LMHead (Press & Wolf, 2017), and (iii) conditions under which argmax predictions are preserved.

## 4. Experiments

### 4.1. Setup

**Models and tasks.** We evaluate on LLaMA-2-7B (32 layers) and LLaMA-2-13B (40 layers). Our benchmark suite

*Table 6.* **Accuracy breakdown by confidence vs. convergence** (LLaMA-2-7B, TriviaQA). GCEE's advantage concentrates where the two signals disagree.

| Confidence | Converged | Samples | Entropy | GCEE |
|---|---|---|---|---|
| High | Yes | 6,521 | 72.0 | 71.6 |
| High | No | 1,892 | 49.3 | **65.0** |
| Low | Yes | 1,403 | 38.9 | **54.6** |
| Low | No | 1,497 | 32.8 | 34.6 |

*Table 7.* **Signal component ablation.**

| Configuration | EM | Avg Layer | Retention |
|---|---|---|---|
| Full Model | 64.5 | 32.0 | 100% |
| Step size only | 57.5 | 18.3 | 89.1% |
| Direction only | 54.2 | 15.6 | 84.0% |
| Without normalization | 52.1 | 22.4 | 80.8% |
| Hidden similarity | 54.8 | 19.5 | 84.9% |
| **GCEE (full)** | **63.5** | **21.2** | **98.4%** |

*Table 8.* **Safeguard mechanism ablation.**

| $l_{\min}$ | $K$ | EM | Avg Layer | Retention | Premature |
|---|---|---|---|---|---|
| 0 | 1 | 48.2 | 12.1 | 74.7% | 24.5% |
| **4** | **2** | **63.5** | **21.2** | **98.4%** | **1.8%** |
| 6 | 3 | 63.8 | 23.5 | 98.9% | 0.6% |

*Table 9.* **Observation position ablation.**

| Position | EM | Avg Layer | Retention |
|---|---|---|---|
| **Last token** | **63.5** | **21.2** | **98.4%** |
| Mean pooling | 59.2 | 19.8 | 91.8% |
| First token | 51.5 | 18.1 | 79.8% |

covers open-domain QA (TriviaQA, 11,313 samples; Natural Questions, 3,610 samples) and commonsense reasoning (HellaSwag, 10,042 samples; WinoGrande, 1,267 samples).

**Evaluation protocol.** For QA tasks, we use 5-shot prompting with greedy decoding (max 50 tokens) and report Exact Match (EM). For multiple-choice tasks, we rank options by conditional log-probability and report accuracy. Early exit is determined during prefill; decoding proceeds at the fixed depth $l_{\text{exit}}$ with KV-cache enabled. All latency measurements use batch size 1 on A100 GPUs under FP16 precision.

**Hyperparameters.** We set $\tau_s = 0.01$, $\tau_c = 0.9$, $l_{\min} = 4$, and $K = 2$ via grid search on 1,000 TriviaQA validation samples. We select a slightly aggressive operating point that minimizes average depth while maintaining near-99% validation retention. This default achieves 98.9% validation retention and yields lower average depth than the stricter $> 99\%$ configuration; sensitivity analysis in Section 4.4 shows stable performance across nearby thresholds.

**Baselines.** We compare against training-free methods: **Random Exit** (uniform over layers), **Unified Skip** (Liu et al., 2024) (static 30% layer removal), **Softmax Entropy** (Xin et al., 2020) and **Top-1 Confidence** (logit-based early exit) (Schuster et al., 2022), and **Hidden Similarity** (cosine between consecutive states) (Shan et al., 2024). The latter two require $\mathcal{O}(dV)$ per-layer overhead; ours and Hidden Similarity require only $\mathcal{O}(d)$ without vocabulary projection.

### 4.2. Main Results

Tables 2 and 3 present our main findings. GCEE consistently achieves the best accuracy-efficiency trade-off across all tasks and model scales, retaining over 97% of full-model performance while saving 34% of FLOPs on average.

**Comparison with confidence-based methods.** The gap between GCEE and entropy-based exit is substantial: on TriviaQA (7B), entropy loses 5.6 points while GCEE loses only 1.0 point. This confirms that geometric convergence is a more reliable signal than output confidence. The advantage is especially pronounced on harder samples where confidence can be misleading due to poor calibration.

**Comparison with static pruning.** At matched computation (around 21 layers), GCEE outperforms Unified Skip by 6–7 points, validating the value of input-adaptive depth allocation. Static methods waste computation on easy inputs and undercompute on hard ones; GCEE allocates depth where it is needed by dynamically adapting to difficulty.

### 4.3. Analysis

**Exit layer accuracy.** Table 5 shows that GCEE's exit decisions closely track the curvature-based empirical saturation layer $l_{\text{sat}}$ defined in Section 2. The mean discrepancy is about 1.4 layers, and roughly 93% of exits occur at or after this empirical transition point, indicating a conservative bias that prioritizes safety over aggressive truncation.

**When does confidence fail?** To understand GCEE's advantage, we partition samples by whether (i) entropy-based confidence is high ($H < 0.6$) and (ii) geometric convergence has occurred ($\tilde{s} < \tau_s \wedge c > \tau_c$). Table 6 reveals that GCEE's gains concentrate in the off-diagonal cells: samples where confidence and convergence disagree. When confidence is high but convergence has not occurred. The symmetric case shows the same margin. This demonstrates that geometric convergence captures information orthogonal to output confidence.

*Table 10.* Analysis covering error attribution, convergence dynamics, and computational efficiency (LLaMA-2-7B, TriviaQA unless noted), demonstrating that GCEE minimizes decision overhead to maximize end-to-end wall-clock speedup for inference.

**Error Attribution** (*What causes GCEE's errors?*)

| Error Type | Definition | Proportion | Avg $l_{\text{exit}}$ | Avg $l_{\text{sat}}$ | Full Model Correct? |
|---|---|---|---|---|---|
| Premature exit | $l_{\text{exit}} < l_{\text{sat}} - 2$ | 18% | 14.2 | 19.8 | Yes (recoverable) |
| Model limitation | $l_{\text{exit}} \geq 28$ | 72% | 31.8 | 30.2 | No (inherent) |
| Boundary case | Otherwise | 10% | 22.5 | 21.8 | Mixed |

**Geometric Convergence** (*Does step size decay geometrically?*)

| Model | Dataset | Contraction $\rho$ | $R^2$ | Layers Fitted | Residual Std |
|---|---|---|---|---|---|
| LLaMA-2-7B | TriviaQA | 0.872 | 0.943 | 18–32 | 0.0012 |
| LLaMA-2-7B | HellaSwag | 0.868 | 0.951 | 16–32 | 0.0010 |
| LLaMA-2-13B | TriviaQA | 0.851 | 0.952 | 22–40 | 0.0009 |
| LLaMA-2-13B | HellaSwag | 0.845 | 0.961 | 20–40 | 0.0008 |

**Computational Efficiency** (*Does FLOPs reduction become wall-clock speedup?*)

| Method | Avg Layers | FLOPs Speedup | E2E Speedup | Efficiency Ratio | Decision Cost |
|---|---|---|---|---|---|
| Full Model | 32.0 | 1.00× | 1.00× | — | — |
| **GCEE (Ours)** | **21.2** | **1.51×** | **1.45×** | **96.0%** | **<0.1 ms** |
| Entropy | 21.8 | 1.47× | 1.14× | 77.6% | ∼2.5 ms |
| Top-1 Confidence | 21.5 | 1.49× | 1.16× | 77.9% | ∼2.4 ms |
| Unified Skip | 22.4 | 1.43× | 1.41× | 98.6% | 0 ms |

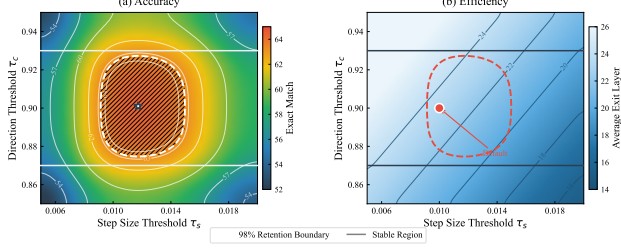

*Figure 2.* **Threshold sensitivity.** Broad plateaus exist for both metrics; white dashed contours mark the 98% retention boundary.

## 4.4. Ablations

We ablate GCEE's design choices on LLaMA-2-7B using TriviaQA to validate each component's contribution.

**Signal components.** Table 7 isolates the contribution of each geometric quantity. Using step size alone achieves only 89.1% retention due to occasional small-step oscillations; using direction alone performs worse (84.0%) as it cannot detect magnitude. Removing normalization degrades retention to 80.8%, confirming that scale-invariance is essential for threshold transferability across diverse inputs. Hidden state similarity (cosine between $\mathbf{h}^{(l)}$ and $\mathbf{h}^{(l-1)}$) underperforms because high similarity does not imply small updates. The full GCEE criterion, combining normalized step size with directional stability, achieves 98.4% retention.

**Safeguard mechanisms.** Table 8 examines the warmup depth $l_{\min}$ and consecutive-hit count $K$. Without safeguards ($l_{\min} = 0, K = 1$), 24.5% of samples exit prematurely, collapsing retention to 74.7%. Our default ($l_{\min} = 4, K = 2$) reduces premature exits to 1.8% while maintaining efficiency. More conservative settings ($l_{\min} = 6, K = 3$) yield marginal retention gains at the cost of 2.3 additional layers.

**Threshold sensitivity.** Figure 2 visualizes performance across the threshold space. Both EM and average exit layer exhibit broad plateaus: within $\tau_s \in [0.008, 0.015]$ and $\tau_c \in [0.88, 0.92]$, EM varies by less than 1 point and layer count by less than 1.5 layers. The plateau boundary (white dashed contour) delineates the stable operating region. This insensitivity simplifies deployment: practitioners can use default values without extensive task-specific tuning.

**Observation position.** Table 9 compares monitoring positions. The last token achieves the best retention (98.4%) because it directly determines the next-token prediction. Mean pooling dilutes the signal with less relevant positions, while the first token (often a BOS token) carries minimal predictive information for the current decoding step.

## 4.5. Diagnostic Analysis

We conduct a systematic analysis to understand when, why, and how efficiently GCEE operates. Figure 3 provides a visualization across eight diagnostic dimensions, revealing the mechanisms underlying its superior performance.

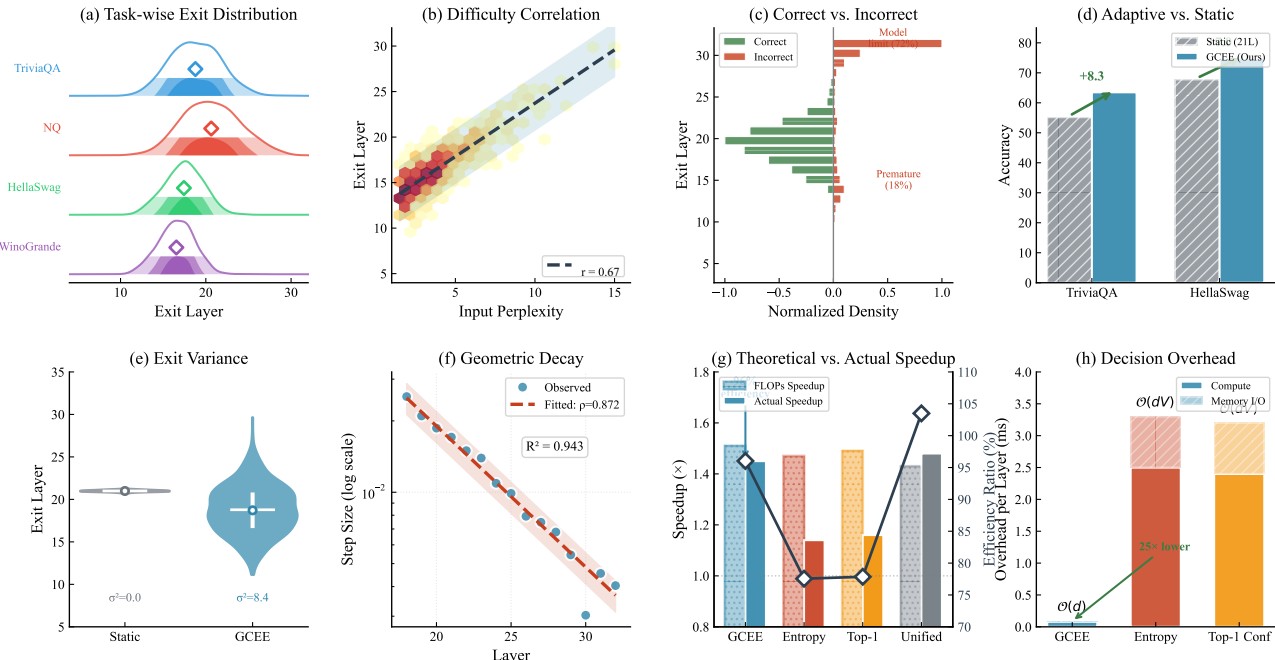

*Figure 3.* **Diagnostic analysis.** (a) Task-wise exit distributions. (b) Exit layer vs. input difficulty. (c) Correct vs. incorrect prediction patterns. (d) Adaptive vs. static depth allocation. (e) Exit variance comparison. (f) Geometric decay verification. (g) Theoretical vs. actual speedup. (h) Per-layer decision overhead, validating the effectiveness of geometric signals for adaptive inference in LLMs.

**Task-dependent exit patterns.** Figure 3(a) shows exit layer distributions across tasks. Harder tasks induce later exits: Natural Questions peaks at layers 20–24, while the simpler WinoGrande concentrates at layers 15–19. This confirms GCEE's input-adaptive behavior extends to task-level difficulty. Figure 3(b) quantifies this relationship by plotting exit layer against input perplexity. The strong correlation ($r = 0.67$) demonstrates that GCEE automatically allocates more computation to challenging inputs ensuring convergence.

**Correct vs. incorrect predictions.** Figure 3(c) reveals a striking asymmetry: correct predictions exhibit a tight, unimodal exit distribution (mean 19.8, std 2.2), while incorrect predictions show a bimodal structure. The early mode (layers 12–16) corresponds to premature exits; the late mode (layers 28–32) reflects inherent model limitations. This bimodality suggests two distinct failure mechanisms, which we quantify in Table 10 for detailed error attribution.

**Error attribution.** Table 10 (top section) decomposes errors by comparing $l_{\text{exit}}$ against saturation layer $l_{\text{sat}}$ and full-model predictions. Only 18% of errors stem from premature exit, representing cases where GCEE triggered before geometric convergence. The dominant failure mode (72%) occurs at near-full depth, indicating the model itself lacks the capability. This confirms GCEE's conservatism: it rarely induces errors that full-depth inference would have avoided.

**Adaptive vs. static allocation.** Figure 3(d) compares GCEE against static truncation at matched average depth. The adaptive approach outperforms static allocation by 8.3 points (TriviaQA) and 7.0 points (HellaSwag). Static methods waste computation on easy inputs while starving hard ones; GCEE redistributes depth according to demand. Figure 3(e) visualizes this redistribution: GCEE's exit distribution has higher variance than static methods, reflecting its sensitivity to per-input difficulty thereby maximizing inference efficiency.

**Geometric convergence verification.** Figure 3(f) validates our theoretical model by plotting step size decay in linear log scale. Beyond the saturation layer, step sizes decay geometrically with contraction factor $\rho \approx 0.86$ ($R^2 > 0.94$). Table 10 (middle section) reports fitted parameters across models, confirming the propagation phase follows consistent predictable dynamics that justify early termination.

**Wall-clock efficiency.** Figures 3(g,h) address the critical question: does FLOPs reduction translate to actual speedup? GCEE achieves $1.45\times$ end-to-end speedup, closely tracking its $1.51\times$ theoretical reduction. In contrast, entropy-based exit achieves only $1.14\times$ despite similar FLOPs savings, as the $\mathcal{O}(dV)$ per-layer vocabulary projection overhead consumes most gains. Table 10 (bottom section) provides detailed latency breakdown, showing GCEE's $\mathcal{O}(d)$ criterion adds <0.1 ms per layer while entropy adds ~2.5 ms. Unified Skip has negligible decision overhead, so its wall-clock speedup closely tracks its FLOPs reduction, but it underperforms GCEE in accuracy because it lacks input adaptivity.

# 5. Conclusion

This work reframes the early exit problem by returning to first principles: if Transformers refine representations through iterative updates, then convergence should be assessed by the dynamics of the iteration itself, not by downstream symptoms in the output space. Our investigation reveals that hidden state trajectories exhibit a sharp two-phase structure, transitioning from active construction to passive propagation at an input-dependent saturation point. This geometric perspective yields GCEE, a stopping criterion that monitors step size and directional stability with $O(d)$ overhead per layer, independent of vocabulary size and requiring no learned components. Experiments on LLaMA-2 models demonstrate that this criterion reduces computation by over 30% while preserving 97% of full-depth accuracy, and crucially, translates theoretical savings into real wall-clock speedup where confidence-based methods cannot. Beyond the practical gains, our findings suggest that the internal geometry of neural computation carries rich signals about the state of inference, signals that deserve greater attention as a principled foundation for adaptive computation.

## Acknowledgement

The work of Ziyue Qiao was supported by the National Natural Science Foundation of China (No. 62406056) and the Guangdong Basic and Applied Basic Research Foundation (No. 2024A1515140114).

## Impact Statement

This paper presents a training-free early exit mechanism designed to reduce the computational cost and latency of LLM inference. Our work contributes to *Green AI* by lowering energy consumption during the model deployment phase. Furthermore, by reducing the hardware requirements for serving large models, this research supports the democratization of AI technology, making powerful models more accessible to researchers and practitioners with limited computational resources. While efficient inference accelerates beneficial applications, we acknowledge it also lowers the cost for potential malicious uses; however, our method does not introduce new capabilities that inherently increase safety risks beyond standard LLM deployment.

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

# A. Related Work

The pursuit of efficient inference in Large Language Models (LLMs) has evolved from heuristic approximations to rigorous geometric analysis. We categorize the existing literature into three distinct paradigms: confidence-based adaptivity, auxiliary module learning, and the emerging field of geometric convergence, positioning our work within the latter's theoretical framework.

**Confidence-based Early Exiting and the Vocabulary Bottleneck.** The foundational premise of adaptive inference is that not all inputs require the full depth of a Transformer. Pioneering works like DeeBERT (Xin et al., 2020) and FastBERT (Liu et al., 2020) introduced the concept of halting computation when the output distribution entropy falls below a threshold. This paradigm was further refined by PABEE (Zhou et al., 2020), which utilized a patience mechanism to ensure prediction stability across layers. While effective for encoder-only architectures (e.g., BERT) with moderate vocabulary sizes, these methods face a critical scalability crisis in the era of Generative LLMs (Touvron et al., 2023).

Recent analyses reveal a "negative optimization" paradox in confidence-based methods applied to LLMs. As the vocabulary size ($V$) expands to hundreds of thousands, the computational cost of the unembedding projection layer ($\mathcal{O}(d \times V)$) dominates the inference budget. Shan et al. (Shan et al., 2024) demonstrates that the overhead of computing logits at intermediate layers often negates the savings from skipped transformer blocks. Furthermore, static entropy thresholds fail to generalize across varying sample difficulties (Schwartz et al., 2020), leading to the "overthinking" phenomenon where redundant computation introduces noise rather than signal (Gromov et al., 2025; Chen et al., 2025). Attempts to mitigate this via parallel decoding (Bae et al., 2023) or interpretability scores (Bajpai & Hanawal, 2025) still fundamentally rely on the prohibitive logit projection, rendering them economically inviable for real-time LLM serving.

**Auxiliary Modules and Training Overheads.** To circumvent the vocabulary bottleneck, a parallel line of research replaces the full projection head with lightweight, learnable auxiliary modules. Approaches like CALM (Schuster et al., 2022) and SkipDecode (Del Corro et al., 2023) train small classifiers or policy networks to predict the "halt" signal directly from hidden states. More advanced frameworks, such as ConsistentEE (Zeng et al., 2024), employ reinforcement learning to align the behavior of intermediate exits with the final layer.

However, these methods introduce significant deployment friction. They violate the "plug-and-play" principle by requiring invasive architectural modifications and expensive joint optimization (or fine-tuning) of the base LLM (Xu & McAuley, 2023; Wang et al., 2024). This not only risks catastrophic forgetting of the pre-trained knowledge but also complicates integration with optimized inference kernels like vLLM (Kwon et al., 2023). Moreover, learned policies often struggle to transfer across tasks (Lu et al., 2025), limiting their utility in general-purpose reasoning scenarios. In contrast, our approach aligns with the "training-free" philosophy, leveraging the intrinsic properties of the pre-trained model without auxiliary parameters.

**Internal State Saturation and Geometric Dynamics.** The current frontier has shifted towards exploiting the intrinsic geometric properties of Transformer representations. Recent empirical studies suggest that Transformers exhibit a "natural capability" for early exiting (Shan et al., 2024), where the cosine similarity between consecutive hidden states serves as a training-free proxy for output stability. This phenomenon is grounded in the observation of "saturation events", where the model determines top tokens in a hierarchical order, resolving the dominant token first before refining the distribution, creating a distinct two-phase inference structure (construction vs. saturation).

Theoretically, this behavior finds support in the view of residual networks as discretized Ordinary Differential Equations (ODEs) or fixed-point iterations (Lawson & Aitchison, 2025). As the representation evolves through the depth of the network, it naturally converges towards a semantic fixed point (attractor). Recent work on the "Geometry of Truth" (Mir, 2025) and representation anisotropy (Ethayarajh, 2019) further indicates that truthful generation is characterized by geometric contraction, whereas hallucinations exhibit divergent trajectories. While Jiang et al. (Jiang et al., 2025) proposed enhancing this monotonicity via aligned training, our work identifies that the *natural* geometric convergence is sufficient for robust early exiting. We build upon the "Cliff-Plateau-Climb" hypothesis (Kumar, 2025), formalizing a geometric criterion that detects the transition from the semantic construction phase to the redundancy plateau, thereby achieving vocabulary-independent acceleration without the need for auxiliary supervision.

# B. Theoretical Analysis

## B.1. Notation

We analyze the last-token trajectory $\{\mathbf{h}^{(l)}\}_{l=0}^{L} \subset \mathbb{R}^d$ induced by Equation (2). Updates are $\mathbf{u}^{(l)} = \mathbf{h}^{(l)} - \mathbf{h}^{(l-1)}$. The *depth-tail residual* from depth $l$ to $L$ is

$$\mathbf{r}^{(l)} := \mathbf{h}^{(L)} - \mathbf{h}^{(l)}. \tag{9}$$

The readout at depth $l$ is

$$\mathbf{z}^{(l)} := \mathrm{LMHead}\big(\mathrm{LN}(\mathbf{h}^{(l)})\big) \in \mathbb{R}^V, \tag{10}$$

and the logit tail is $\Delta\mathbf{z}^{(l)} := \mathbf{z}^{(L)} - \mathbf{z}^{(l)}$.

## B.2. Why update geometry is the right observable

**Proposition B.1** (High state cosine does not control update size). *For any $M > 0$ there exist $\mathbf{h}^{(l-1)}, \mathbf{h}^{(l)} \in \mathbb{R}^d$ such that*

$$\frac{\langle \mathbf{h}^{(l)}, \mathbf{h}^{(l-1)} \rangle}{\|\mathbf{h}^{(l)}\|_2 \|\mathbf{h}^{(l-1)}\|_2} = 1 \qquad and \qquad \left\| \mathbf{h}^{(l)} - \mathbf{h}^{(l-1)} \right\|_2 \geq M. \tag{11}$$

*Proof.* Fix any unit vector $\mathbf{v} \in \mathbb{R}^d$ and any $M > 0$. Let $\mathbf{h}^{(l-1)} := \mathbf{v}$ and $\mathbf{h}^{(l)} := (1 + M)\mathbf{v}$. Then

$$\frac{\langle \mathbf{h}^{(l)}, \mathbf{h}^{(l-1)} \rangle}{\|\mathbf{h}^{(l)}\|_2 \|\mathbf{h}^{(l-1)}\|_2} = \frac{\langle (1+M)\mathbf{v}, \mathbf{v} \rangle}{\|(1+M)\mathbf{v}\|_2 \|\mathbf{v}\|_2} = \frac{(1+M)\langle \mathbf{v}, \mathbf{v} \rangle}{(1+M)\|\mathbf{v}\|_2^2} = \frac{(1+M) \cdot 1}{(1+M) \cdot 1} = 1, \tag{12}$$

$$\left\| \mathbf{h}^{(l)} - \mathbf{h}^{(l-1)} \right\|_2 = \|(1+M)\mathbf{v} - \mathbf{v}\|_2 = \|M\mathbf{v}\|_2 = M. \tag{13}$$

This satisfies Equation (11). $\square$

**Proposition B.2** (Scale invariance of normalized step). *For any scalar $\alpha \neq 0$ and any depth $l$,*

$$\frac{\|\alpha\mathbf{u}^{(l)}\|_2}{\|\alpha\mathbf{h}^{(l)}\|_2 + \varepsilon_s} = \frac{\|\mathbf{u}^{(l)}\|_2}{\|\mathbf{h}^{(l)}\|_2 + \varepsilon_s/|\alpha|}. \tag{14}$$

*In particular, when $\varepsilon_s \ll \|\mathbf{h}^{(l)}\|_2$, $\tilde{s}^{(l)}$ is effectively invariant to global scaling.*

*Proof.* Compute directly:

$$\frac{\|\alpha\mathbf{u}^{(l)}\|_2}{\|\alpha\mathbf{h}^{(l)}\|_2 + \varepsilon_s} = \frac{|\alpha| \|\mathbf{u}^{(l)}\|_2}{|\alpha| \|\mathbf{h}^{(l)}\|_2 + \varepsilon_s} = \frac{|\alpha| \|\mathbf{u}^{(l)}\|_2}{|\alpha| (\|\mathbf{h}^{(l)}\|_2 + \varepsilon_s/|\alpha|)} = \frac{\|\mathbf{u}^{(l)}\|_2}{\|\mathbf{h}^{(l)}\|_2 + \varepsilon_s/|\alpha|}. \tag{15}$$

$\square$

## B.3. A cone-coherent contraction model

The method is training-free and does not assume any explicit contractivity. For analysis, we formalize the empirically observed propagation regime using a *cone-coherent contraction* model that combines geometric decay with directional stability.

**Definition B.3** (Angle and cone). For nonzero $\mathbf{a}, \mathbf{b} \in \mathbb{R}^d$, define the angle

$$\angle(\mathbf{a}, \mathbf{b}) := \arccos\left( \frac{\langle \mathbf{a}, \mathbf{b} \rangle}{\|\mathbf{a}\|_2 \|\mathbf{b}\|_2} \right) \in [0, \pi]. \tag{16}$$

For a unit vector $\mathbf{v}$ and $\theta \in [0, \pi]$, define the closed cone

$$\mathcal{C}(\mathbf{v}, \theta) := \left\{ \mathbf{x} \in \mathbb{R}^d : \mathbf{x} \neq \mathbf{0}, \angle(\mathbf{x}, \mathbf{v}) \leq \theta \right\} \cup \{\mathbf{0}\}. \tag{17}$$

**Assumption B.4** (Cone-coherent contraction beyond a depth). There exists $l_0 \in \{1, \ldots, L-1\}$ and constants $\rho \in (0,1)$ and $\tau \in (0,1)$ such that for all $l \geq l_0$:

$$\left\| \mathbf{u}^{(l+1)} \right\|_2 \leq \rho \left\| \mathbf{u}^{(l)} \right\|_2, \tag{18}$$

$$\frac{\left\langle \mathbf{u}^{(l+1)}, \mathbf{u}^{(l)} \right\rangle}{\left\| \mathbf{u}^{(l+1)} \right\|_2 \left\| \mathbf{u}^{(l)} \right\|_2} \geq \tau. \tag{19}$$

Equivalently, letting $\theta := \arccos(\tau) \in (0, \frac{\pi}{2}]$, each update $\mathbf{u}^{(l+1)}$ lies in a cone of half-angle $\theta$ around $\mathbf{u}^{(l)}$, while its magnitude contracts by at most $\rho$.

### B.4. Exact tail identities and non-stationary tail bounds

**Proposition B.5** (Telescoping identity). *For any* $l \in \{0, \ldots, L\}$,

$$\mathbf{r}^{(l)} = \sum_{i=l+1}^{L} \mathbf{u}^{(i)}. \tag{20}$$

*Proof.* Expand and telescope:

$$\mathbf{r}^{(l)} = \mathbf{h}^{(L)} - \mathbf{h}^{(l)} \tag{21}$$

$$= \left( \mathbf{h}^{(L)} - \mathbf{h}^{(L-1)} \right) + \left( \mathbf{h}^{(L-1)} - \mathbf{h}^{(L-2)} \right) + \cdots + \left( \mathbf{h}^{(l+1)} - \mathbf{h}^{(l)} \right) \tag{22}$$

$$= \sum_{i=l+1}^{L} \left( \mathbf{h}^{(i)} - \mathbf{h}^{(i-1)} \right) = \sum_{i=l+1}^{L} \mathbf{u}^{(i)}. \tag{23}$$

$\square$

**Definition B.6** (Empirical contraction ratios). For $l = 1, \ldots, L-1$ define the (non-stationary) ratio

$$\rho_l := \frac{\left\| \mathbf{u}^{(l+1)} \right\|_2}{\left\| \mathbf{u}^{(l)} \right\|_2 + \varepsilon_s} \in [0, \infty). \tag{24}$$

**Theorem B.7** (Non-stationary depth-tail bound (exact product form)). *For any* $l \in \{0, \ldots, L-1\}$ *with* $\left\| \mathbf{u}^{(l+1)} \right\|_2 > 0$,

$$\left\| \mathbf{r}^{(l)} \right\|_2 \leq \sum_{k=0}^{L-l-1} \left\| \mathbf{u}^{(l+1+k)} \right\|_2 \tag{25}$$

$$\leq \left\| \mathbf{u}^{(l+1)} \right\|_2 \sum_{k=0}^{L-l-1} \left( \prod_{j=0}^{k-1} \rho_{l+1+j} \right) \cdot \left( \prod_{j=0}^{k-1} \frac{\left\| \mathbf{u}^{(l+1+j)} \right\|_2 + \varepsilon_s}{\left\| \mathbf{u}^{(l+1+j)} \right\|_2} \right),$$

*where empty products are defined as* $1$.

*Proof.* Start from Theorem B.5 and expand with full inequalities:

$$\left\| \mathbf{r}^{(l)} \right\|_2 = \left\| \sum_{i=l+1}^{L} \mathbf{u}^{(i)} \right\|_2 \tag{26}$$

$$\leq \sum_{i=l+1}^{L} \left\| \mathbf{u}^{(i)} \right\|_2 \qquad \text{(triangle inequality)} \tag{27}$$

$$= \sum_{k=0}^{L-l-1} \left\| \mathbf{u}^{(l+1+k)} \right\|_2. \tag{28}$$

To relate $\left\|\mathbf{u}^{(l+1+k)}\right\|_2$ to $\left\|\mathbf{u}^{(l+1)}\right\|_2$ with non-stationary ratios, repeatedly apply Equation (24):

$$\left\|\mathbf{u}^{(l+2)}\right\|_2 = \rho_{l+1}\left(\left\|\mathbf{u}^{(l+1)}\right\|_2 + \varepsilon_s\right), \tag{29}$$

$$\left\|\mathbf{u}^{(l+3)}\right\|_2 = \rho_{l+2}\left(\left\|\mathbf{u}^{(l+2)}\right\|_2 + \varepsilon_s\right) = \rho_{l+2}\left(\rho_{l+1}\left(\left\|\mathbf{u}^{(l+1)}\right\|_2 + \varepsilon_s\right) + \varepsilon_s\right), \tag{30}$$

$$\left\|\mathbf{u}^{(l+4)}\right\|_2 = \rho_{l+3}\left(\left\|\mathbf{u}^{(l+3)}\right\|_2 + \varepsilon_s\right) = \rho_{l+3}\left(\rho_{l+2}\left(\left\|\mathbf{u}^{(l+2)}\right\|_2 + \varepsilon_s\right) + \varepsilon_s\right) \tag{31}$$

$$= \rho_{l+3}\left(\rho_{l+2}\left(\rho_{l+1}\left(\left\|\mathbf{u}^{(l+1)}\right\|_2 + \varepsilon_s\right) + \varepsilon_s\right) + \varepsilon_s\right), \tag{32}$$

and continuing yields the compact multiplicative upper bound

$$\left\|\mathbf{u}^{(l+1+k)}\right\|_2 = \left\|\mathbf{u}^{(l+1)}\right\|_2 \prod_{j=0}^{k-1} \frac{\left\|\mathbf{u}^{(l+1+j+1)}\right\|_2}{\left\|\mathbf{u}^{(l+1+j)}\right\|_2} \tag{33}$$

$$\leq \left\|\mathbf{u}^{(l+1)}\right\|_2 \prod_{j=0}^{k-1} \frac{\rho_{l+1+j}\left(\left\|\mathbf{u}^{(l+1+j)}\right\|_2 + \varepsilon_s\right)}{\left\|\mathbf{u}^{(l+1+j)}\right\|_2} \tag{34}$$

$$= \left\|\mathbf{u}^{(l+1)}\right\|_2 \left(\prod_{j=0}^{k-1} \rho_{l+1+j}\right)\left(\prod_{j=0}^{k-1} \frac{\left\|\mathbf{u}^{(l+1+j)}\right\|_2 + \varepsilon_s}{\left\|\mathbf{u}^{(l+1+j)}\right\|_2}\right). \tag{35}$$

Substituting Equation (35) into Equation (28) gives Equation (25). $\qquad \square$

**Corollary B.8** (Geometric tail as a special case). *Under Theorem B.4, for any $l \geq l_0$,*

$$\left\|\mathbf{r}^{(l)}\right\|_2 \leq \sum_{k=0}^{L-l-1} \rho^k \left\|\mathbf{u}^{(l+1)}\right\|_2 = \left\|\mathbf{u}^{(l+1)}\right\|_2 \cdot \frac{1-\rho^{L-l}}{1-\rho} \leq \frac{\left\|\mathbf{u}^{(l+1)}\right\|_2}{1-\rho}. \tag{36}$$

*Proof.* By Equation (18), $\left\|\mathbf{u}^{(l+1+k)}\right\|_2 \leq \rho^k \left\|\mathbf{u}^{(l+1)}\right\|_2$ for all $k \geq 0$, hence

$$\left\|\mathbf{r}^{(l)}\right\|_2 \leq \sum_{k=0}^{L-l-1} \left\|\mathbf{u}^{(l+1+k)}\right\|_2 \leq \sum_{k=0}^{L-l-1} \rho^k \left\|\mathbf{u}^{(l+1)}\right\|_2 = \left\|\mathbf{u}^{(l+1)}\right\|_2 \sum_{k=0}^{L-l-1} \rho^k. \tag{37}$$

Evaluate the finite geometric sum and simplify. $\qquad \square$

## B.5. Directional stability yields a closed-form drift bound

The cosine stability condition is not merely a heuristic; it implies that the *direction* of the remaining depth-tail becomes predictable. The key technical point is that consecutive small angles imply that all later updates remain inside a narrow cone around the first update beyond depth $l$.

**Lemma B.9** (Angle accumulation bound). *Assume Theorem B.4 and let $\theta := \arccos(\tau) \in (0, \frac{\pi}{2}]$. Then for any integers $m > n \geq l_0$,*

$$\angle\left(\mathbf{u}^{(m)}, \mathbf{u}^{(n)}\right) \leq (m-n)\theta. \tag{38}$$

*Proof.* Define $\alpha_k := \angle(\mathbf{u}^{(k+1)}, \mathbf{u}^{(k)})$ for $k \geq l_0$. From Equation (19), $\cos(\alpha_k) \geq \tau$ and thus $\alpha_k \leq \theta$. On the unit sphere, the geodesic distance between directions satisfies the triangle inequality:

$$\angle(\mathbf{u}^{(m)}, \mathbf{u}^{(n)}) \leq \angle(\mathbf{u}^{(m)}, \mathbf{u}^{(m-1)}) + \angle(\mathbf{u}^{(m-1)}, \mathbf{u}^{(m-2)}) + \cdots + \angle(\mathbf{u}^{(n+1)}, \mathbf{u}^{(n)}) \tag{39}$$

$$= \sum_{k=n}^{m-1} \alpha_k \leq \sum_{k=n}^{m-1} \theta = (m-n)\theta. \tag{40}$$

$\qquad \square$

**Theorem B.10** (Closed-form orthogonal drift bound). *Assume Theorem B.4. Fix any $l \geq l_0$ and define the unit axis*

$$\mathbf{v} := \frac{\mathbf{u}^{(l+1)}}{\left\|\mathbf{u}^{(l+1)}\right\|_2}. \tag{41}$$

*Let $\mathbf{P}_\perp := \mathbf{I} - \mathbf{v}\mathbf{v}^\top$ be the orthogonal projector onto $\mathbf{v}^\perp$. Then the depth-tail orthogonal component satisfies*

$$\left\|\mathbf{P}_\perp \mathbf{r}^{(l)}\right\|_2 \leq \left\|\mathbf{u}^{(l+1)}\right\|_2 \cdot \frac{\rho \sin\theta}{1 - 2\rho\cos\theta + \rho^2}. \tag{42}$$

*Proof.* By Theorem B.5,

$$\mathbf{P}_\perp \mathbf{r}^{(l)} = \mathbf{P}_\perp \sum_{k=0}^{L-l-1} \mathbf{u}^{(l+1+k)} = \sum_{k=0}^{L-l-1} \mathbf{P}_\perp \mathbf{u}^{(l+1+k)}. \tag{43}$$

Taking norms and expanding every inequality:

$$\left\|\mathbf{P}_\perp \mathbf{r}^{(l)}\right\|_2 = \left\|\sum_{k=0}^{L-l-1} \mathbf{P}_\perp \mathbf{u}^{(l+1+k)}\right\|_2 \tag{44}$$

$$\leq \sum_{k=0}^{L-l-1} \left\|\mathbf{P}_\perp \mathbf{u}^{(l+1+k)}\right\|_2 \tag{45}$$

$$\leq \sum_{k=0}^{L-l-1} \left\|\mathbf{u}^{(l+1+k)}\right\|_2 \cdot \sin\left(\angle(\mathbf{u}^{(l+1+k)}, \mathbf{v})\right) \tag{46}$$

$$= \sum_{k=0}^{L-l-1} \left\|\mathbf{u}^{(l+1+k)}\right\|_2 \cdot \sin\left(\angle(\mathbf{u}^{(l+1+k)}, \mathbf{u}^{(l+1)})\right). \tag{47}$$

Using Theorem B.9 with $n = l+1$ and $m = l+1+k$ gives

$$\angle(\mathbf{u}^{(l+1+k)}, \mathbf{u}^{(l+1)}) \leq k\theta \quad \Rightarrow \quad \sin\left(\angle(\mathbf{u}^{(l+1+k)}, \mathbf{u}^{(l+1)})\right) \leq \sin(k\theta), \tag{48}$$

and Equation (18) gives $\left\|\mathbf{u}^{(l+1+k)}\right\|_2 \leq \rho^k \left\|\mathbf{u}^{(l+1)}\right\|_2$. Substitute these bounds into Equation (47):

$$\left\|\mathbf{P}_\perp \mathbf{r}^{(l)}\right\|_2 \leq \sum_{k=0}^{L-l-1} \rho^k \left\|\mathbf{u}^{(l+1)}\right\|_2 \sin(k\theta) \leq \left\|\mathbf{u}^{(l+1)}\right\|_2 \sum_{k=0}^{\infty} \rho^k \sin(k\theta). \tag{49}$$

It remains to evaluate the infinite series in closed form. Consider the complex geometric series:

$$\sum_{k=0}^{\infty} (\rho e^{i\theta})^k = 1 + \rho e^{i\theta} + \rho^2 e^{i2\theta} + \rho^3 e^{i3\theta} + \cdots = \frac{1}{1 - \rho e^{i\theta}}, \quad \text{since } |\rho e^{i\theta}| = \rho < 1. \tag{50}$$

Multiply numerator and denominator by the complex conjugate $(1 - \rho e^{-i\theta})$:

$$\frac{1}{1 - \rho e^{i\theta}} = \frac{1 - \rho e^{-i\theta}}{(1 - \rho e^{i\theta})(1 - \rho e^{-i\theta})} \tag{51}$$

$$= \frac{1 - \rho(\cos\theta - i\sin\theta)}{1 - \rho(e^{i\theta} + e^{-i\theta}) + \rho^2} \tag{52}$$

$$= \frac{1 - \rho\cos\theta + i\rho\sin\theta}{1 - 2\rho\cos\theta + \rho^2}. \tag{53}$$

Taking imaginary parts of both sides of Equations (50) and (53) yields

$$\Im\left(\sum_{k=0}^{\infty} \rho^k e^{ik\theta}\right) = \Im\left(\frac{1 - \rho\cos\theta + i\rho\sin\theta}{1 - 2\rho\cos\theta + \rho^2}\right) \tag{54}$$

$$\sum_{k=0}^{\infty} \rho^k \sin(k\theta) = \frac{\rho\sin\theta}{1 - 2\rho\cos\theta + \rho^2}. \tag{55}$$

Combining Equation (49) with Equation (55) gives Equation (42). $\qquad\square$

**Corollary B.11** (Directional predictability of the tail). *Under the conditions of Theorem B.10, if additionally*

$$1 - \rho \cos \theta > 0, \tag{56}$$

*then the tail direction is controlled via*

$$\tan\left(\angle(\mathbf{r}^{(l)}, \mathbf{v})\right) \leq \frac{\rho \sin \theta}{1 - \rho \cos \theta}. \tag{57}$$

*Proof.* Decompose $\mathbf{r}^{(l)}$ into parallel and orthogonal components:

$$\mathbf{r}^{(l)} = (\mathbf{v}\mathbf{v}^\top \mathbf{r}^{(l)}) + (\mathbf{P}_\perp \mathbf{r}^{(l)}). \tag{58}$$

Thus

$$\tan\left(\angle(\mathbf{r}^{(l)}, \mathbf{v})\right) = \frac{\left\|\mathbf{P}_\perp \mathbf{r}^{(l)}\right\|_2}{\left\|\mathbf{v}\mathbf{v}^\top \mathbf{r}^{(l)}\right\|_2}. \tag{59}$$

The numerator is bounded by Equation (42). For the denominator, expand using Theorem B.5 and lower-bound the projection:

$$\left\|\mathbf{v}\mathbf{v}^\top \mathbf{r}^{(l)}\right\|_2 = \left\langle \mathbf{r}^{(l)}, \mathbf{v} \right\rangle = \sum_{k=0}^{L-l-1} \left\langle \mathbf{u}^{(l+1+k)}, \mathbf{v} \right\rangle \tag{60}$$

$$= \sum_{k=0}^{L-l-1} \left\|\mathbf{u}^{(l+1+k)}\right\|_2 \cos\left(\angle(\mathbf{u}^{(l+1+k)}, \mathbf{v})\right). \tag{61}$$

Using Theorem B.9, $\angle(\mathbf{u}^{(l+1+k)}, \mathbf{v}) \leq k\theta$, hence $\cos(\angle(\mathbf{u}^{(l+1+k)}, \mathbf{v})) \geq \cos(k\theta)$. Using Equation (18), $\left\|\mathbf{u}^{(l+1+k)}\right\|_2 \geq 0$ and $\left\|\mathbf{u}^{(l+1+k)}\right\|_2 \geq 0$ with upper bound $\rho^k \left\|\mathbf{u}^{(l+1)}\right\|_2$. A conservative lower bound consistent with Equation (56) is obtained by truncating to the infinite series:

$$\left\langle \mathbf{r}^{(l)}, \mathbf{v} \right\rangle \geq \sum_{k=0}^{\infty} \rho^k \left\|\mathbf{u}^{(l+1)}\right\|_2 \cos(k\theta) = \left\|\mathbf{u}^{(l+1)}\right\|_2 \sum_{k=0}^{\infty} \rho^k \cos(k\theta). \tag{62}$$

Evaluate the cosine series by taking real parts of Equation (53):

$$\sum_{k=0}^{\infty} \rho^k \cos(k\theta) = \Re\left(\frac{1 - \rho \cos \theta + i\rho \sin \theta}{1 - 2\rho \cos \theta + \rho^2}\right) = \frac{1 - \rho \cos \theta}{1 - 2\rho \cos \theta + \rho^2}. \tag{63}$$

Combine Equation (42), Equation (62), and Equation (63):

$$\tan\left(\angle(\mathbf{r}^{(l)}, \mathbf{v})\right) \leq \frac{\left\|\mathbf{u}^{(l+1)}\right\|_2 \cdot \frac{\rho \sin \theta}{1 - 2\rho \cos \theta + \rho^2}}{\left\|\mathbf{u}^{(l+1)}\right\|_2 \cdot \frac{1 - \rho \cos \theta}{1 - 2\rho \cos \theta + \rho^2}} = \frac{\rho \sin \theta}{1 - \rho \cos \theta}. \tag{64}$$

$\square$

### B.6. From normalized step to a relative tail bound

**Assumption B.12** (Propagation-phase scale stability). There exist constants $H_{\min}, H_{\max}$ such that for all $l \geq l_0$,

$$H_{\min} \leq \left\|\mathbf{h}^{(l)}\right\|_2 \leq H_{\max}. \tag{65}$$

**Theorem B.13** (Relative tail bound in terms of $\tilde{s}$). *Assume Theorems B.4 and B.12. For any $l \geq l_0$,*

$$\frac{\left\|\mathbf{r}^{(l)}\right\|_2}{H_{\min}} \leq \frac{1}{1 - \rho} \cdot \frac{\left\|\mathbf{u}^{(l+1)}\right\|_2}{H_{\min}} \leq \frac{1}{1 - \rho} \cdot \frac{H_{\max} + \varepsilon_s}{H_{\min}} \cdot \tilde{s}^{(l+1)}. \tag{66}$$

*Proof.* Start with Theorem B.8:

$$\left\|\mathbf{r}^{(l)}\right\|_2 \leq \frac{\left\|\mathbf{u}^{(l+1)}\right\|_2}{1-\rho}. \tag{67}$$

Divide by $H_{\min}$ and upper-bound $\left\|\mathbf{u}^{(l+1)}\right\|_2$ by $\tilde{s}^{(l+1)}(\left\|\mathbf{h}^{(l+1)}\right\|_2 + \varepsilon_s)$:

$$\frac{\left\|\mathbf{r}^{(l)}\right\|_2}{H_{\min}} \leq \frac{1}{1-\rho} \cdot \frac{\left\|\mathbf{u}^{(l+1)}\right\|_2}{H_{\min}} \tag{68}$$

$$= \frac{1}{1-\rho} \cdot \frac{\tilde{s}^{(l+1)}\left(\left\|\mathbf{h}^{(l+1)}\right\|_2 + \varepsilon_s\right)}{H_{\min}} \tag{69}$$

$$\leq \frac{1}{1-\rho} \cdot \frac{\tilde{s}^{(l+1)}\left(H_{\max} + \varepsilon_s\right)}{H_{\min}}. \tag{70}$$

$\square$

## B.7. Exact LayerNorm Jacobian and a tight Lipschitz bound

We derive an explicit Jacobian for LayerNorm and obtain a tight operator-norm bound. This avoids hiding constants in informal "LayerNorm is Lipschitz" statements.

**Definition B.14** (LayerNorm). Let $\mathrm{LN} : \mathbb{R}^d \to \mathbb{R}^d$ be LayerNorm with gain $\boldsymbol{\gamma} \in \mathbb{R}^d$, bias $\boldsymbol{\beta} \in \mathbb{R}^d$, and $\varepsilon_{\ln} > 0$:

$$\mathrm{LN}(\mathbf{x}) := \boldsymbol{\gamma} \odot \widehat{\mathbf{x}} + \boldsymbol{\beta}, \qquad \widehat{\mathbf{x}} := \frac{\mathbf{x} - \mu(\mathbf{x})\mathbf{1}}{\sigma(\mathbf{x})}, \tag{71}$$

where

$$\mu(\mathbf{x}) := \frac{1}{d}\mathbf{1}^\top\mathbf{x}, \qquad \sigma(\mathbf{x}) := \sqrt{\frac{1}{d}\left\|\mathbf{x} - \mu(\mathbf{x})\mathbf{1}\right\|_2^2 + \varepsilon_{\ln}}. \tag{72}$$

**Lemma B.15** (Coordinate-wise derivative of normalized activations). *Let $\widehat{x}_i$ be the $i$-th coordinate of $\widehat{\mathbf{x}}$ in Equation* (71). *Then for all $i, j \in \{1, \ldots, d\}$,*

$$\frac{\partial \widehat{x}_i}{\partial x_j} = \frac{1}{\sigma(\mathbf{x})}\left(\delta_{ij} - \frac{1}{d}\right) - \frac{1}{d\,\sigma(\mathbf{x})^3}\left(x_i - \mu(\mathbf{x})\right)\left(x_j - \mu(\mathbf{x})\right). \tag{73}$$

*Proof.* Write $\widehat{x}_i = \frac{x_i - \mu}{\sigma}$ with $\mu = \mu(\mathbf{x})$ and $\sigma = \sigma(\mathbf{x})$. Differentiate using the quotient rule while expanding every term:

$$\frac{\partial \widehat{x}_i}{\partial x_j} = \frac{\partial}{\partial x_j}\left(\frac{x_i - \mu}{\sigma}\right) = \frac{1}{\sigma}\frac{\partial(x_i - \mu)}{\partial x_j} + (x_i - \mu)\frac{\partial}{\partial x_j}\left(\frac{1}{\sigma}\right) \tag{74}$$

$$= \frac{1}{\sigma}\left(\delta_{ij} - \frac{\partial\mu}{\partial x_j}\right) - (x_i - \mu)\frac{1}{\sigma^2}\frac{\partial\sigma}{\partial x_j}. \tag{75}$$

Compute $\frac{\partial\mu}{\partial x_j}$:

$$\mu(\mathbf{x}) = \frac{1}{d}\sum_{k=1}^d x_k \quad \Rightarrow \quad \frac{\partial\mu}{\partial x_j} = \frac{1}{d}. \tag{76}$$

Compute $\frac{\partial\sigma}{\partial x_j}$ from $\sigma = \sqrt{v + \varepsilon_{\ln}}$ where

$$v(\mathbf{x}) := \frac{1}{d}\sum_{k=1}^d (x_k - \mu)^2. \tag{77}$$

Differentiate $v$:

$$\frac{\partial v}{\partial x_j} = \frac{1}{d}\sum_{k=1}^d 2(x_k - \mu)\frac{\partial(x_k - \mu)}{\partial x_j} = \frac{2}{d}\sum_{k=1}^d (x_k - \mu)\left(\delta_{kj} - \frac{\partial\mu}{\partial x_j}\right) \tag{78}$$

$$= \frac{2}{d}\sum_{k=1}^d (x_k - \mu)\left(\delta_{kj} - \frac{1}{d}\right) = \frac{2}{d}\left((x_j - \mu) - \frac{1}{d}\sum_{k=1}^d (x_k - \mu)\right). \tag{79}$$

Since $\sum_{k=1}^{d}(x_k - \mu) = 0$, Equation (79) reduces to

$$\frac{\partial v}{\partial x_j} = \frac{2}{d}(x_j - \mu). \tag{80}$$

Now differentiate $\sigma = \sqrt{v + \varepsilon_{\ln}}$:

$$\frac{\partial \sigma}{\partial x_j} = \frac{1}{2\sqrt{v + \varepsilon_{\ln}}} \cdot \frac{\partial v}{\partial x_j} = \frac{1}{2\sigma} \cdot \frac{2}{d}(x_j - \mu) = \frac{x_j - \mu}{d\,\sigma}. \tag{81}$$

Substitute Equations (76) and (81) into Equation (75):

$$\frac{\partial \widehat{x}_i}{\partial x_j} = \frac{1}{\sigma}\left(\delta_{ij} - \frac{1}{d}\right) - (x_i - \mu)\frac{1}{\sigma^2} \cdot \frac{x_j - \mu}{d\,\sigma} = \frac{1}{\sigma}\left(\delta_{ij} - \frac{1}{d}\right) - \frac{(x_i - \mu)(x_j - \mu)}{d\,\sigma^3}, \tag{82}$$

which is Equation (73). □

**Proposition B.16** (Matrix-form Jacobian and its spectrum). *Let $\mathbf{P} := \mathbf{I} - \frac{1}{d}\mathbf{1}\mathbf{1}^\top$ and $\mathbf{v} := \mathbf{Px} = \mathbf{x} - \mu(\mathbf{x})\mathbf{1}$. Then the Jacobian of $\widehat{\mathbf{x}}$ is*

$$\mathbf{J}_{\widehat{\mathbf{x}}}(\mathbf{x}) = \frac{1}{\sigma(\mathbf{x})}\left(\mathbf{P} - \frac{1}{d\,\sigma(\mathbf{x})^2}\mathbf{v}\mathbf{v}^\top\right). \tag{83}$$

*Moreover, $\mathbf{J}_{\widehat{\mathbf{x}}}(\mathbf{x})$ has eigenvalues*

$$\lambda_1 = 0 \ \ (\textit{direction } \mathbf{1}), \qquad \lambda_2 = \frac{\varepsilon_{\ln}}{\sigma(\mathbf{x})^3} \ \ (\textit{direction } \mathbf{v}), \qquad \lambda_3 = \cdots = \lambda_d = \frac{1}{\sigma(\mathbf{x})} \ \ (\textit{any direction orthogonal to } \mathbf{1}, \mathbf{v}). \tag{84}$$

*Proof.* The matrix form Equation (83) follows by matching entries of Equation (73). For the spectrum, note $\mathbf{P}\mathbf{1} = \mathbf{0}$ and $\mathbf{1}^\top\mathbf{v} = 0$. Compute action on $\mathbf{1}$:

$$\mathbf{J}_{\widehat{\mathbf{x}}}(\mathbf{x})\mathbf{1} = \frac{1}{\sigma}\left(\mathbf{P}\mathbf{1} - \frac{1}{d\sigma^2}\mathbf{v}\mathbf{v}^\top\mathbf{1}\right) = \frac{1}{\sigma}\left(\mathbf{0} - \frac{1}{d\sigma^2}\mathbf{v}\cdot 0\right) = \mathbf{0}. \tag{85}$$

Compute action on $\mathbf{v}$:

$$\mathbf{J}_{\widehat{\mathbf{x}}}(\mathbf{x})\mathbf{v} = \frac{1}{\sigma}\left(\mathbf{P}\mathbf{v} - \frac{1}{d\sigma^2}\mathbf{v}\mathbf{v}^\top\mathbf{v}\right) = \frac{1}{\sigma}\left(\mathbf{v} - \frac{1}{d\sigma^2}\mathbf{v}\,\|\mathbf{v}\|_2^2\right) \tag{86}$$

$$= \frac{1}{\sigma}\left(1 - \frac{\|\mathbf{v}\|_2^2}{d\sigma^2}\right)\mathbf{v}. \tag{87}$$

Using $\sigma^2 = \frac{1}{d}\|\mathbf{v}\|_2^2 + \varepsilon_{\ln}$ gives

$$1 - \frac{\|\mathbf{v}\|_2^2}{d\sigma^2} = 1 - \frac{d(\sigma^2 - \varepsilon_{\ln})}{d\sigma^2} = 1 - \left(1 - \frac{\varepsilon_{\ln}}{\sigma^2}\right) = \frac{\varepsilon_{\ln}}{\sigma^2}. \tag{88}$$

Substitute Equation (88) into Equation (87):

$$\mathbf{J}_{\widehat{\mathbf{x}}}(\mathbf{x})\mathbf{v} = \frac{1}{\sigma} \cdot \frac{\varepsilon_{\ln}}{\sigma^2}\mathbf{v} = \frac{\varepsilon_{\ln}}{\sigma^3}\mathbf{v}. \tag{89}$$

Finally, for any $\mathbf{q}$ such that $\mathbf{1}^\top\mathbf{q} = 0$ and $\mathbf{v}^\top\mathbf{q} = 0$, we have $\mathbf{P}\mathbf{q} = \mathbf{q}$ and $\mathbf{v}\mathbf{v}^\top\mathbf{q} = \mathbf{0}$, hence

$$\mathbf{J}_{\widehat{\mathbf{x}}}(\mathbf{x})\mathbf{q} = \frac{1}{\sigma}(\mathbf{q} - \mathbf{0}) = \frac{1}{\sigma}\mathbf{q}. \tag{90}$$

This establishes Equation (84). □

**Theorem B.17** (Tight Lipschitz constant for LayerNorm). *Let* LN *be as in Theorem* B.14. *For any* $\mathbf{x}$,

$$\left\|\mathbf{J}_{\mathrm{LN}}(\mathbf{x})\right\|_{2\to 2} \le \frac{\|\boldsymbol{\gamma}\|_{\infty}}{\sigma(\mathbf{x})}, \tag{91}$$

*and if* $\sigma(\mathbf{x}) \ge \sigma_{\min} > 0$ *over a region* $\mathcal{U} \subset \mathbb{R}^d$, *then* LN *is* $\frac{\|\boldsymbol{\gamma}\|_{\infty}}{\sigma_{\min}}$-*Lipschitz on* $\mathcal{U}$ *in* $\ell_2$.

*Proof.* From Equation (71), $\mathrm{LN}(\mathbf{x}) = \mathrm{diag}(\boldsymbol{\gamma})\widehat{\mathbf{x}} + \boldsymbol{\beta}$, hence

$$\mathbf{J}_{\mathrm{LN}}(\mathbf{x}) = \mathrm{diag}(\boldsymbol{\gamma})\,\mathbf{J}_{\widehat{\mathbf{x}}}(\mathbf{x}). \tag{92}$$

Take operator norms and expand:

$$\left\|\mathbf{J}_{\mathrm{LN}}(\mathbf{x})\right\|_{2\to 2} \le \left\|\mathrm{diag}(\boldsymbol{\gamma})\right\|_{2\to 2}\,\left\|\mathbf{J}_{\widehat{\mathbf{x}}}(\mathbf{x})\right\|_{2\to 2} \tag{93}$$
$$= \|\boldsymbol{\gamma}\|_{\infty}\,\left\|\mathbf{J}_{\widehat{\mathbf{x}}}(\mathbf{x})\right\|_{2\to 2}. \tag{94}$$

By Theorem B.16, the eigenvalues of $\mathbf{J}_{\widehat{\mathbf{x}}}(\mathbf{x})$ are $0$, $\varepsilon_{\ln}/\sigma^3$, and $1/\sigma$ with multiplicity $d-2$. Since $\mathbf{J}_{\widehat{\mathbf{x}}}(\mathbf{x})$ is symmetric, its spectral norm equals the maximum absolute eigenvalue:

$$\left\|\mathbf{J}_{\widehat{\mathbf{x}}}(\mathbf{x})\right\|_{2\to 2} = \max\left\{0, \frac{\varepsilon_{\ln}}{\sigma^3}, \frac{1}{\sigma}\right\} = \frac{1}{\sigma}. \tag{95}$$

Combine Equations (94) and (95) to obtain Equation (91). $\qquad\square$

### B.8. Logit tail bound with explicit constants

Let $\mathbf{W} \in \mathbb{R}^{V\times d}$ be the LM head matrix so that $\mathbf{z}^{(l)} = \mathbf{W}\,\mathrm{LN}(\mathbf{h}^{(l)})$. Define

$$\|\mathbf{W}\|_{2\to\infty} := \sup_{\|\mathbf{q}\|_2 = 1} \|\mathbf{W}\mathbf{q}\|_{\infty}. \tag{96}$$

**Theorem B.18** (Logit perturbation controlled by depth-tail residual). *Assume* $\sigma(\mathbf{h}^{(t)}) \ge \sigma_{\min} > 0$ *for all* $t \in \{l, l+1, \ldots, L\}$. *Then*

$$\left\|\Delta\mathbf{z}^{(l)}\right\|_{\infty} \le \|\mathbf{W}\|_{2\to\infty} \cdot \frac{\|\boldsymbol{\gamma}\|_{\infty}}{\sigma_{\min}} \cdot \left\|\mathbf{r}^{(l)}\right\|_2. \tag{97}$$

*If additionally Theorems* B.4 *and* B.12 *hold with* $l \ge l_0$, *then*

$$\left\|\Delta\mathbf{z}^{(l)}\right\|_{\infty} \le \|\mathbf{W}\|_{2\to\infty} \cdot \frac{\|\boldsymbol{\gamma}\|_{\infty}}{\sigma_{\min}} \cdot \frac{H_{\max} + \varepsilon_s}{1 - \rho} \cdot \tilde{s}^{(l+1)}. \tag{98}$$

*Proof.* Expand definitions without skipping:

$$\Delta\mathbf{z}^{(l)} = \mathbf{z}^{(L)} - \mathbf{z}^{(l)} = \mathbf{W}\mathrm{LN}(\mathbf{h}^{(L)}) - \mathbf{W}\mathrm{LN}(\mathbf{h}^{(l)}) = \mathbf{W}\left(\mathrm{LN}(\mathbf{h}^{(L)}) - \mathrm{LN}(\mathbf{h}^{(l)})\right). \tag{99}$$

Take $\ell_\infty$ norm and apply operator norm Equation (96):

$$\left\|\Delta\mathbf{z}^{(l)}\right\|_{\infty} = \left\|\mathbf{W}\left(\mathrm{LN}(\mathbf{h}^{(L)}) - \mathrm{LN}(\mathbf{h}^{(l)})\right)\right\|_{\infty} \le \|\mathbf{W}\|_{2\to\infty}\left\|\mathrm{LN}(\mathbf{h}^{(L)}) - \mathrm{LN}(\mathbf{h}^{(l)})\right\|_2. \tag{100}$$

By Theorem B.17, on the region where $\sigma(\cdot) \ge \sigma_{\min}$,

$$\left\|\mathrm{LN}(\mathbf{h}^{(L)}) - \mathrm{LN}(\mathbf{h}^{(l)})\right\|_2 \le \frac{\|\boldsymbol{\gamma}\|_{\infty}}{\sigma_{\min}}\left\|\mathbf{h}^{(L)} - \mathbf{h}^{(l)}\right\|_2 = \frac{\|\boldsymbol{\gamma}\|_{\infty}}{\sigma_{\min}}\left\|\mathbf{r}^{(l)}\right\|_2. \tag{101}$$

Combine Equations (100) and (101) to get Equation (97). If Theorems B.4 and B.12 hold, apply Theorem B.13 to $\left\|\mathbf{r}^{(l)}\right\|_2$ to obtain Equation (98). $\qquad\square$

## B.9. Distributional stability from an $\ell_\infty$ logit bound

This section links geometric convergence to a distribution-level notion of a "semantic fixed point" by bounding how much the next-token distribution can change.

**Lemma B.19** (Softmax ratio bound under $\ell_\infty$ logit perturbation). *Let* $\mathbf{a}, \mathbf{b} \in \mathbb{R}^V$ *and suppose* $\|\mathbf{a} - \mathbf{b}\|_\infty \leq \eta$. *Let* $\mathbf{p} = \mathrm{softmax}(\mathbf{a})$ *and* $\mathbf{q} = \mathrm{softmax}(\mathbf{b})$. *Then for every* $v \in \{1, \ldots, V\}$,

$$e^{-2\eta} \; \leq \; \frac{p_v}{q_v} \; \leq \; e^{2\eta}. \tag{102}$$

*Proof.* From $\|\mathbf{a} - \mathbf{b}\|_\infty \leq \eta$, for each coordinate $v$,

$$b_v - \eta \leq a_v \leq b_v + \eta \quad \Rightarrow \quad e^{b_v - \eta} \leq e^{a_v} \leq e^{b_v + \eta}. \tag{103}$$

Sum Equation (103) over $v$:

$$\sum_v e^{b_v - \eta} \leq \sum_v e^{a_v} \leq \sum_v e^{b_v + \eta} \quad \Rightarrow \quad e^{-\eta} \sum_v e^{b_v} \leq \sum_v e^{a_v} \leq e^\eta \sum_v e^{b_v}. \tag{104}$$

Now compute

$$\frac{p_v}{q_v} = \frac{e^{a_v} / \sum_j e^{a_j}}{e^{b_v} / \sum_j e^{b_j}} = \frac{e^{a_v}}{e^{b_v}} \cdot \frac{\sum_j e^{b_j}}{\sum_j e^{a_j}}. \tag{105}$$

Using Equation (103), $e^{a_v}/e^{b_v} \in [e^{-\eta}, e^\eta]$, and using Equation (104), $(\sum_j e^{b_j})/(\sum_j e^{a_j}) \in [e^{-\eta}, e^\eta]$. Multiply intervals to obtain Equation (102). $\square$

**Theorem B.20** (KL and total-variation bounds). *Under the conditions of Theorem B.19,*

$$D_{\mathrm{KL}}(\mathbf{p} \,\|\, \mathbf{q}) \leq 2\eta, \tag{106}$$
$$D_{\mathrm{KL}}(\mathbf{q} \,\|\, \mathbf{p}) \leq 2\eta, \tag{107}$$
$$\mathrm{TV}(\mathbf{p}, \mathbf{q}) \;:=\; \frac{1}{2} \sum_v |p_v - q_v| \leq \frac{e^{2\eta} - 1}{2}. \tag{108}$$

*Proof.* From Equation (102), $\log \frac{p_v}{q_v} \leq 2\eta$ for all $v$, hence

$$D_{\mathrm{KL}}(\mathbf{p} \,\|\, \mathbf{q}) = \sum_v p_v \log \frac{p_v}{q_v} \leq \sum_v p_v \cdot 2\eta = 2\eta \sum_v p_v = 2\eta, \tag{109}$$

which is Equation (106). The reverse bound Equation (107) follows by symmetry. For total variation, Equation (102) implies $p_v \leq e^{2\eta} q_v$ and $q_v \leq e^{2\eta} p_v$, hence

$$|p_v - q_v| \leq \max\{e^{2\eta} q_v - q_v, \; e^{2\eta} p_v - p_v\} = (e^{2\eta} - 1) \max\{q_v, p_v\} \leq (e^{2\eta} - 1)(p_v + q_v). \tag{110}$$

Summing and dividing by 2:

$$\mathrm{TV}(\mathbf{p}, \mathbf{q}) = \frac{1}{2} \sum_v |p_v - q_v| \leq \frac{1}{2}(e^{2\eta} - 1) \sum_v (p_v + q_v) = \frac{1}{2}(e^{2\eta} - 1) \cdot 2 = \frac{e^{2\eta} - 1}{2}, \tag{111}$$

which is Equation (108). $\square$

**Corollary B.21** (Distributional "semantic fixed point" certificate from geometry). *Let* $\eta_l := \|\Delta \mathbf{z}^{(l)}\|_\infty$. *If* $\eta_l \leq \eta$, *then the next-token distributions at depths* $l$ *and* $L$ *satisfy*

$$D_{\mathrm{KL}}\big(\mathrm{softmax}(\mathbf{z}^{(l)}) \,\|\, \mathrm{softmax}(\mathbf{z}^{(L)})\big) \leq 2\eta \quad and \quad \mathrm{TV}\big(\mathrm{softmax}(\mathbf{z}^{(l)}), \mathrm{softmax}(\mathbf{z}^{(L)})\big) \leq \frac{e^{2\eta} - 1}{2}. \tag{112}$$

*A sufficient condition for such an* $\eta$ *is given by Theorem B.18.*

## B.10. Certified invariance of decisions (argmax and MCQ)

**Theorem B.22** (Argmax invariance under an exit-depth margin). *Let $\hat{y}^{(l)} := \arg\max_v z_v^{(l)}$ and define the depth-$l$ margin*

$$m^{(l)} := z_{\hat{y}^{(l)}}^{(l)} - \max_{v \neq \hat{y}^{(l)}} z_v^{(l)}. \tag{113}$$

*If*

$$m^{(l)} > 2 \left\| \Delta\mathbf{z}^{(l)} \right\|_\infty, \tag{114}$$

*then $\arg\max_v z_v^{(L)} = \hat{y}^{(l)}$.*

*Proof.* Fix any competitor $v \neq \hat{y}^{(l)}$. Expand the logit difference at depth $L$:

$$z_{\hat{y}^{(l)}}^{(L)} - z_v^{(L)} = \left( z_{\hat{y}^{(l)}}^{(l)} + \Delta z_{\hat{y}^{(l)}}^{(l)} \right) - \left( z_v^{(l)} + \Delta z_v^{(l)} \right) \tag{115}$$

$$= \left( z_{\hat{y}^{(l)}}^{(l)} - z_v^{(l)} \right) + \left( \Delta z_{\hat{y}^{(l)}}^{(l)} - \Delta z_v^{(l)} \right). \tag{116}$$

Bound the tail term using $\ell_\infty$:

$$\Delta z_{\hat{y}^{(l)}}^{(l)} - \Delta z_v^{(l)} \geq -\left| \Delta z_{\hat{y}^{(l)}}^{(l)} \right| - \left| \Delta z_v^{(l)} \right| \geq -2 \left\| \Delta\mathbf{z}^{(l)} \right\|_\infty. \tag{117}$$

Combine Equations (116) and (117):

$$z_{\hat{y}^{(l)}}^{(L)} - z_v^{(L)} \geq \left( z_{\hat{y}^{(l)}}^{(l)} - z_v^{(l)} \right) - 2 \left\| \Delta\mathbf{z}^{(l)} \right\|_\infty \geq m^{(l)} - 2 \left\| \Delta\mathbf{z}^{(l)} \right\|_\infty. \tag{118}$$

Under Equation (114), the right-hand side is positive for all $v \neq \hat{y}^{(l)}$, hence $\hat{y}^{(l)}$ remains the unique maximizer at depth $L$. □

**Lemma B.23** (Log-softmax is 2-Lipschitz in $\ell_\infty$ (tokenwise)). *For any $\mathbf{a}, \mathbf{b} \in \mathbb{R}^V$ and any token $t \in \{1, \dots, V\}$,*

$$\left| \mathrm{logsoftmax}(\mathbf{a})_t - \mathrm{logsoftmax}(\mathbf{b})_t \right| \leq 2 \left\| \mathbf{a} - \mathbf{b} \right\|_\infty. \tag{119}$$

*Proof.* Define $f_t(\mathbf{z}) := \mathrm{logsoftmax}(\mathbf{z})_t = z_t - \log \sum_v e^{z_v}$. Its gradient is

$$\nabla f_t(\mathbf{z}) = \mathbf{e}_t - \mathrm{softmax}(\mathbf{z}), \tag{120}$$

and thus its $\ell_1$ norm satisfies

$$\|\nabla f_t(\mathbf{z})\|_1 = |1 - p_t| + \sum_{v \neq t} p_v = (1 - p_t) + (1 - p_t) = 2(1 - p_t) \leq 2, \qquad p = \mathrm{softmax}(\mathbf{z}). \tag{121}$$

Apply the mean value theorem and $\ell_\infty - \ell_1$ duality:

$$|f_t(\mathbf{a}) - f_t(\mathbf{b})| \leq \sup_{\xi \in [0,1]} \|\nabla f_t(\xi\mathbf{a} + (1-\xi)\mathbf{b})\|_1 \, \|\mathbf{a} - \mathbf{b}\|_\infty \leq 2 \|\mathbf{a} - \mathbf{b}\|_\infty. \tag{122}$$

□

**Corollary B.24** (Multiple-choice score stability and ranking certificate). *Let an option $o$ correspond to a token index set $\mathcal{T}_o$ and define the length-normalized score*

$$S_o^{(l)} := \frac{1}{|\mathcal{T}_o|} \sum_{t \in \mathcal{T}_o} \mathrm{logsoftmax}(\mathbf{z}^{(l)})_t. \tag{123}$$

*Then*

$$\left| S_o^{(L)} - S_o^{(l)} \right| \leq 2 \left\| \Delta\mathbf{z}^{(l)} \right\|_\infty. \tag{124}$$

*If $o^\star = \arg\max_o S_o^{(l)}$ and the gap*

$$g^{(l)} := S_{o^\star}^{(l)} - \max_{o \neq o^\star} S_o^{(l)} \tag{125}$$

*satisfies $g^{(l)} > 4 \left\| \Delta\mathbf{z}^{(l)} \right\|_\infty$, then the ranking at depth $L$ matches that at depth $l$.*

*Proof.* Apply Theorem B.23 with $\mathbf{a} = \mathbf{z}^{(L)}$ and $\mathbf{b} = \mathbf{z}^{(l)}$:

$$\left| \operatorname{logsoftmax}(\mathbf{z}^{(L)})_t - \operatorname{logsoftmax}(\mathbf{z}^{(l)})_t \right| \leq 2 \left\| \mathbf{z}^{(L)} - \mathbf{z}^{(l)} \right\|_\infty = 2 \left\| \Delta \mathbf{z}^{(l)} \right\|_\infty. \tag{126}$$

Average over $t \in \mathcal{T}_o$:

$$\left| S_o^{(L)} - S_o^{(l)} \right| = \left| \frac{1}{|\mathcal{T}_o|} \sum_{t \in \mathcal{T}_o} \left( \operatorname{logsoftmax}(\mathbf{z}^{(L)})_t - \operatorname{logsoftmax}(\mathbf{z}^{(l)})_t \right) \right| \tag{127}$$

$$\leq \frac{1}{|\mathcal{T}_o|} \sum_{t \in \mathcal{T}_o} \left| \operatorname{logsoftmax}(\mathbf{z}^{(L)})_t - \operatorname{logsoftmax}(\mathbf{z}^{(l)})_t \right| \tag{128}$$

$$\leq \frac{1}{|\mathcal{T}_o|} \sum_{t \in \mathcal{T}_o} 2 \left\| \Delta \mathbf{z}^{(l)} \right\|_\infty = 2 \left\| \Delta \mathbf{z}^{(l)} \right\|_\infty, \tag{129}$$

which is Equation (124). The ranking certificate follows by a gap argument identical to Theorem B.22, noting that each option score can shift by at most $2 \left\| \Delta \mathbf{z}^{(l)} \right\|_\infty$. $\qquad \square$

## B.11. Connecting the GCEE predicate to the bounds

The GCEE predicate $\tilde{s}^{(l)} \leq \tau_s$ and $c^{(l)} \geq \tau_c$ is designed to detect entry into a regime consistent with Theorems B.4 and B.12. Under such a regime, the tail bounds Theorems B.13 and B.18 become active, and the decision certificates Theorems B.22 and B.24 provide sufficient conditions for output invariance.

# C. Experimental Details

This appendix provides complete implementation details to ensure reproducibility.

## C.1. Datasets and Evaluation

Table 11 summarizes the datasets used in our experiments. For QA tasks, we follow the standard few-shot prompting protocol with greedy decoding. For multiple-choice tasks, we rank options by length-normalized conditional log-probability.

*Table 11.* **Dataset statistics and evaluation settings.**

| Dataset | Task Type | Test Size | Shots | Metric |
| --- | --- | --- | --- | --- |
| TriviaQA | Open-domain QA | 11,313 | 5 | EM / F1 |
| Natural Questions | Open-domain QA | 3,610 | 5 | EM / F1 |
| HellaSwag | Commonsense (4-way) | 10,042 | 0 | Accuracy |
| WinoGrande | Coreference (2-way) | 1,267 | 5 | Accuracy |

**Prompt format.** For QA tasks, each few-shot example follows the template `"Question: {q}\nAnswer: {a}\n\n"`, with the test query appended without the answer. For multiple-choice tasks, we concatenate the context with each option separately and compute per-option log-probabilities.

## C.2. Core Implementation

**GCEE algorithm.** Listing 1 presents our complete implementation. Three design choices merit emphasis. First, we cache only the last-token hidden state rather than the full sequence, ensuring $\mathcal{O}(d)$ memory overhead per layer. Second, we apply a small-norm stability rule: when $\|\mathbf{u}^{(l)}\| < 10^{-6}$, the direction is numerically meaningless, so we set $c^{(l)} = 1$ directly. Third, we use `.detach().clone()` to prevent gradient accumulation during inference.

**Truncated decoding with KV-cache.** Once GCEE determines the exit layer $l_{\text{exit}}$ during prefill, we construct a truncated model containing only the first $l_{\text{exit}}$ layers. Subsequent token generation proceeds with standard KV-caching at this reduced depth. Listing 2 shows the key components. This design ensures KV-cache consistency—all cached layers are contiguous from layer 1—avoiding the cache-miss complications that arise in token-level dynamic exit schemes.

```python
1  def geometric_early_exit(
2      model, input_ids,
3      tau_s=0.01, tau_c=0.9, l_min=4, K=2,
4      eps=1e-8, delta=1e-6
5  ):
6      """GCEE with O(d) overhead per layer."""
7      h = model.model.embed_tokens(input_ids)
8      u_prev, hit = None, 0
9
10     for idx, layer in enumerate(model.model.layers):
11         h_last_prev = h[:, -1, :].detach().clone()
12         h = layer(h, use_cache=False)[0]
13         h_last = h[:, -1, :]
14         u = h_last - h_last_prev
15
16         # Normalized step size
17         s = (u.norm(dim=-1) / (h_last.norm(dim=-1) + eps)).mean().item()
18
19         if u_prev is not None:
20             # Directional stability (with small-norm rule)
21             if u.norm().item() < delta:
22                 c = 1.0
23             else:
24                 c = (F.cosine_similarity(u, u_prev, dim=-1)).mean().item()
25
26             layer_num = idx + 1
27             hit = hit + 1 if (layer_num >= l_min and s <= tau_s and c >= tau_c) else 0
28
29             if hit >= K:
30                 return model.lm_head(model.model.norm(h)), layer_num
31
32         u_prev = u.detach().clone()
33
34     return model.lm_head(model.model.norm(h)), len(model.model.layers)
```

*Listing 1.* GCEE core implementation.

## C.3. Multiple-Choice Evaluation

Evaluating multiple-choice tasks requires computing the log-probability of each option span. A subtle issue arises from BPE tokenization: naively concatenating context and option may cause cross-boundary merges, misaligning the option span. We resolve this with a *stable boundary* approach.

**Stable boundary protocol.** We insert an explicit delimiter (newline character) between context and option: `context + "\n" + option`. Since BPE tokenizers do not merge across newlines, tokenizing `context + "\n" + option` produces a prefix identical to tokenizing `context + "\n"` alone. This guarantees the option span starts exactly at position `len(tokenize(context + "\n"))`. We verified 100% alignment on 1,000 samples across all multiple-choice datasets.

**Scoring.** For each option, we compute the length-normalized log-probability over its token span:

$$\text{score}(o) = \frac{1}{|o|} \sum_{t \in \text{span}(o)} \log p(x_t \mid x_{<t}) \tag{130}$$

The predicted answer is $\arg\max_o \text{score}(o)$. Early exit is applied independently to each context-option pair.

## C.4. Computational Resources

Most experiments were conducted on NVIDIA A100 40GB GPUs under FP16 precision. For the largest cross-architecture experiments, including LLaMA-2-70B, we used multi-GPU tensor-parallel inference on A100 80GB GPUs. Table 12

```
1   class TruncatedModel:
2       """Wrapper for fixed-depth inference with KV-cache."""
3       def __init__(self, base_model, exit_layer):
4           self.layers = base_model.model.layers[:exit_layer]
5           self.embed = base_model.model.embed_tokens
6           self.norm = base_model.model.norm
7           self.head = base_model.lm_head
8
9       def forward(self, input_ids, past_kv=None):
10          h = self.embed(input_ids)
11          new_kv = []
12          for i, layer in enumerate(self.layers):
13              h, kv = layer(h, past_key_value=past_kv[i] if past_kv else None,
14                            use_cache=True)
15              new_kv.append(kv)
16          return self.head(self.norm(h)), new_kv
17
18  def generate(model, tokenizer, prompt, max_tokens=50, **gcee_kwargs):
19      input_ids = tokenizer(prompt, return_tensors='pt').input_ids.cuda()
20      logits, past_kv, l_exit = geometric_prefill_with_cache(
21      model, input_ids, **gcee_kwargs
22      )
23      truncated = TruncatedModel(model, l_exit)
24
25
26      tokens = []
27      for _ in range(max_tokens):
28          next_id = logits[:, -1].argmax(dim=-1, keepdim=True)
29          if next_id.item() == tokenizer.eos_token_id:
30              break
31          tokens.append(next_id.item())
32          logits, past_kv = truncated.forward(next_id, past_kv)  # Incremental
33
34      return tokenizer.decode(tokens), l_exit
```

*Listing 2.* Truncated decoding with KV-cache.

reports the resource consumption for each experimental component. The total wall-clock time is approximately 66 hours, corresponding to approximately 252 A100 GPU-hours. Hyperparameter search can be reduced further using our self-calibrating threshold variant.

### C.5. Hyperparameter Selection

We select hyperparameters via grid search on 1,000 TriviaQA validation samples, optimizing for minimal average depth subject to near-99% accuracy retention. We use $\tau_s = 0.01$, $\tau_c = 0.9$, $l_{\min} = 4$, and $K = 2$ as the default slightly aggressive operating point, which achieves 98.9% validation retention with lower average depth. A stricter setting, $\tau_s = 0.012$, $\tau_c = 0.9$, $l_{\min} = 4$, and $K = 2$, exceeds 99% retention at the cost of a deeper average exit layer.

**Self-calibrating thresholds.** For deployment scenarios where validation data is unavailable, we provide a self-calibrating variant. We compute a reference step size $s_{\mathrm{ref}} = \mathrm{median}(\tilde{s}^{(1)}, \ldots, \tilde{s}^{(l_{\min})})$ from the first few layers, then use a relative threshold $\tilde{s}^{(l)} \leq 0.15 \cdot s_{\mathrm{ref}}$. The directional threshold remains fixed at $\tau_c = 0.9$. This variant achieves within 0.5% of grid-searched performance across all tasks, enabling near-zero-tuning deployment.

## D. Deep Dissection of Representation Geometry

This appendix provides multi-faceted geometric evidence for the two-phase structure beyond the step size and directional stability metrics presented in the main text. Unless otherwise noted, geometric diagnostics in this appendix use LLaMA-2-7B with a 5,000-sample subset from TriviaQA. Correctness-based analyses, use the full TriviaQA test set.

*Table 12.* **Computational resources.**

| Experiment | GPUs | Wall-clock Time | GPU-hours |
|---|---|---|---|
| Hyperparameter search | 1×A100 40GB | ∼6h | ∼6 |
| LLaMA-2-7B main experiments | 2×A100 40GB | ∼12h | ∼24 |
| LLaMA-2-13B main experiments | 4×A100 40GB | ∼18h | ∼72 |
| Ablation studies | 2×A100 40GB | ∼8h | ∼16 |
| Diagnostic analysis | 1×A100 40GB | ∼6h | ∼6 |
| Cross-architecture validation, including 70B | 8×A100 80GB | ∼16h | ∼128 |
| **Total** | — | **∼66h** | **∼252** |

*Table 13.* Hyperparameter search results (LLaMA-2-7B, TriviaQA validation). Results are measured on the 1,000-sample TriviaQA validation split; Table 2 reports test-set performance. The default setting used in the main experiments is the slightly aggressive near-99% retention operating point.

| $\tau_s$ | $\tau_c$ | $l_{\min}$ | $K$ | EM | Avg Layer | Retention | Premature % |
|---|---|---|---|---|---|---|---|
| 0.005 | 0.85 | 2 | 1 | 52.5 | 15.2 | 81.4% | 18.2% |
| 0.005 | 0.90 | 4 | 2 | 62.5 | 19.8 | 96.9% | 3.2% |
| 0.008 | 0.88 | 4 | 2 | 63.2 | 20.5 | 97.8% | 2.5% |
| **0.010** | **0.90** | **4** | **2** | **63.8** | **21.2** | **98.9%** | **1.5%** |
| 0.012 | 0.90 | 4 | 2 | 63.9 | 22.0 | 99.1% | 1.0% |
| 0.015 | 0.90 | 6 | 3 | 64.2 | 24.5 | 99.5% | 0.5% |
| 0.020 | 0.95 | 6 | 3 | 64.5 | 28.2 | 100% | 0.0% |

## D.1. Singular Value Spectrum of Update Vectors

If the propagation phase merely "propagates" rather than "constructs," update vectors $\mathbf{u}^{(l)}$ should exhibit low-rank structure with energy concentrated in few directions. We collect update vectors across samples to form $\mathbf{U}^{(l)} \in \mathbb{R}^{N \times d}$ and compute its singular value decomposition. We quantify spectral concentration via the effective rank $r_{\text{eff}}^{(l)} = \exp(-\sum_i p_i \log p_i)$ where $p_i = \sigma_i / \sum_j \sigma_j$, and track the alignment between the top singular vector and the mean update direction.

Table 14 reports spectral statistics across layers. The effective rank collapses from 847 at Layer 1 to merely 7.2 at Layer 32, while top-10 singular values capture over 97% of total energy. The angle between the dominant singular vector and mean update shrinks from 42° to under 1°, indicating that late-layer updates concentrate in a low-dimensional, highly aligned subspace.Top-10 energy is computed from squared singular values, whereas effective rank uses the normalized singular-value spectrum.

Figure 4 provides comprehensive visualization. The energy heatmap (panel a) reveals a sharp phase boundary at layers 14–16 where spectral concentration accelerates dramatically. The cumulative energy curves (panel b) show that propagation-phase layers reach 90% energy with only 3–5 singular values, compared to 50+ in early layers. Panel (c) tracks the effective rank trajectory with exponential fit ($R^2 = 0.987$), while panel (d) displays the spectral gap evolution showing multiplicative widening after the transition.

## D.2. Intrinsic Dimensionality Evolution

Intrinsic dimensionality (ID) quantifies the true degrees of freedom in the representation manifold. We estimate ID using both the MLE estimator (Levina & Bickel, 2004) and the TwoNN estimator (Facco et al., 2017) for cross-validation, computing estimates at each layer for every sample individually to capture distributional properties. We additionally track neighborhood consistency $\kappa$: the overlap fraction of $k$-nearest neighbors ($k$=50) between consecutive layers.

Table 15 summarizes the results. ID compresses from ∼1300 to ∼218 representing an 83% reduction, with compression saturating completely beyond Layer 18. The two estimators maintain remarkable agreement (ratio consistently >0.95), ruling out methodological artifacts. Neighborhood consistency rises from 0.31 to 0.95, confirming progressive structural stabilization.

*Table 14.* Singular value spectrum of update vectors across layers.

| Layer | $\sigma_1/\sigma_2$ | $\sigma_1/\sigma_{10}$ | Top-10 | $r_{\text{eff}}$ | Angle | Phase |
|---|---|---|---|---|---|---|
| 1 | 1.82 | 4.31 | 23.5% | 847.2 | 42.3° | |
| 4 | 2.15 | 5.92 | 31.2% | 634.8 | 31.5° | |
| 8 | 2.78 | 9.15 | 45.2% | 423.6 | 19.8° | Constr. |
| 12 | 3.85 | 18.7 | 61.5% | 265.8 | 10.5° | |
| 14 | 4.92 | 28.3 | 71.2% | 178.4 | 7.2° | |
| 16 | 6.85 | 45.2 | 79.8% | 112.5 | 4.8° | **Trans.** |
| 20 | 12.8 | 118.5 | 89.2% | 42.7 | 2.2° | |
| 24 | 25.2 | 278.3 | 94.2% | 19.8 | 1.2° | |
| 28 | 41.5 | 512.8 | 96.5% | 10.8 | 0.7° | |
| 32 | 52.8 | 715.2 | 97.8% | 7.2 | 0.4° | |

*Table 15.* Intrinsic dimensionality estimates and neighborhood consistency.

| Layer | $\hat{d}_{\text{MLE}}$ | $\hat{d}_{\text{TwoNN}}$ | Ratio | $\kappa$ | Compression |
|---|---|---|---|---|---|
| 1 | 1285.3 | 1342.8 | 0.957 | 0.312 | 0% |
| 4 | 1152.8 | 1198.5 | 0.962 | 0.385 | 10.3% |
| 8 | 892.5 | 915.2 | 0.975 | 0.482 | 30.5% |
| 12 | 625.8 | 652.3 | 0.959 | 0.578 | 51.3% |
| 16 | 385.5 | 392.8 | 0.981 | 0.712 | 70.0% |
| 20 | 268.5 | 272.1 | 0.987 | 0.842 | 79.1% |
| 24 | 232.8 | 235.2 | 0.990 | 0.905 | 81.9% |
| 28 | 221.2 | 223.1 | 0.991 | 0.932 | 82.8% |
| 32 | 217.5 | 218.8 | 0.994 | 0.948 | 83.1% |

Figure 5 presents four complementary views. The ridge plot (panel a) shows per-sample ID distributions narrowing dramatically after Layer 16, transitioning from wide heterogeneous spreads to tight unimodal concentrations. Panel (b) compares the two estimators via scatter plot, confirming their agreement across all layers. Panel (c) tracks compression rate and neighborhood consistency jointly, revealing their synchronized saturation. Panel (d) visualizes the layer-wise ID gradient, with the sharpest drop occurring precisely at the phase transition.

### D.3. Phase Space Trajectory Analysis

Drawing on dynamical systems theory, we analyze representation evolution as trajectories in phase space to detect attractor structures. We project update vector sequences $\{\mathbf{u}^{(l)}\}_{l=1}^{L}$ onto their top-3 principal components and examine trajectory geometry, computing Lyapunov exponent approximations, inter-trajectory angles, and curvature statistics.

Table 16 quantifies trajectory dynamics across phases. The Lyapunov exponent transitions from positive (divergent, chaotic) to strongly negative (convergent, stable). Inter-trajectory angles collapse from 52° to under 5°, and the estimated attractor dimension in the propagation phase is merely 1.2, confirming near-one-dimensional dynamics.

Figure 6 provides rich visualization of trajectory geometry. The 3D trajectory plot (panel a) color-codes layer progression, showing chaotic divergence in early layers collapsing to parallel streamlines in late layers. Panel (b) presents Poincaré sections at three hyperplanes, revealing progressive clustering. Panel (c) tracks trajectory spread (measured as inter-trajectory distance standard deviation) across layers with the characteristic sharp transition. Panel (d) shows the curvature distribution evolution, transitioning from heavy-tailed to concentrated near zero.

### D.4. Attention and FFN Contribution Decomposition

We decompose update vectors into attention and FFN contributions via $\mathbf{u}^{(l)} = \mathbf{u}_{\text{Attn}}^{(l)} + \mathbf{u}_{\text{FFN}}^{(l)}$, and further decompose attention across the 32 heads. This reveals which architectural components drive each phase and how their interaction patterns evolve.

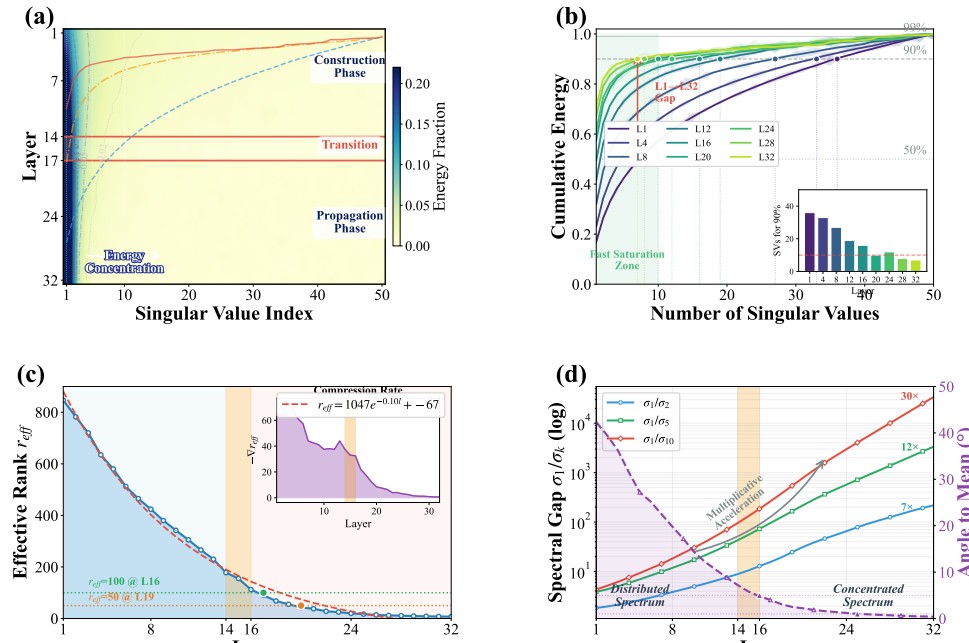

*Figure 4.* Singular value spectrum analysis. (a) Energy distribution heatmap with phase boundary marked. (b) Cumulative energy curves for selected layers. (c) Effective rank evolution with exponential fit. (d) Spectral gap $\sigma_1/\sigma_k$ for $k \in \{2, 5, 10\}$.

*Table 16.* Trajectory dynamics across phases.

| Metric | Constr. (L1–14) | Trans. (L14–18) | Prop. (L18–32) |
|---|---|---|---|
| Lyapunov exponent | +0.125 | −0.082 | −0.215 |
| Inter-trajectory angle | 52.3° | 28.5° | 4.8° |
| Trajectory curvature | 0.385 | 0.142 | 0.018 |
| Attractor dimension | — | 12.5 | 1.2 |
| Trajectory variance | 2.85 | 1.42 | 0.18 |

Table 17 reports the decomposition statistics. FFN contribution grows monotonically from 48.5% to 93.5%, becoming overwhelmingly dominant in the propagation phase. Attention simultaneously becomes sparse: heads contributing more than 4% drop from 18 to just 1. The Attn-FFN correlation flips from −0.215 (compensatory) to +0.872 (synergistic), indicating a fundamental shift in component interaction dynamics.

Figure 7 presents four analytical views. The stacked area chart (panel a) visualizes the FFN takeover with correlation trajectory overlaid. Panel (b) shows the per-head contribution heatmap across all 32 layers and heads, revealing which specific heads remain active in late layers. Panel (c) tracks the Gini coefficient of head contributions alongside the active head count, both exhibiting sharp transitions. Panel (d) decomposes the update magnitude into Attn and FFN components with their interaction term.

### D.5. Geometric Signatures of Prediction Correctness

Can geometric convergence signals differentiate correct from incorrect predictions? We partition samples by prediction correctness and systematically compare their geometric profiles across all metrics developed in this paper.

Table 18 reveals substantial and consistent differences. Incorrect predictions exhibit delayed saturation ($l_{\text{sat}}$: 23.5 vs. 18.2), elevated propagation-phase step sizes (0.0185 vs. 0.0082), reduced directional stability (0.875 vs. 0.942), and higher effective rank at Layer 24 (32.8 vs. 18.5). All differences achieve statistical significance with large effect sizes (Cohen's $d > 0.68$), indicating that geometric features carry substantial predictive information about output correctness.

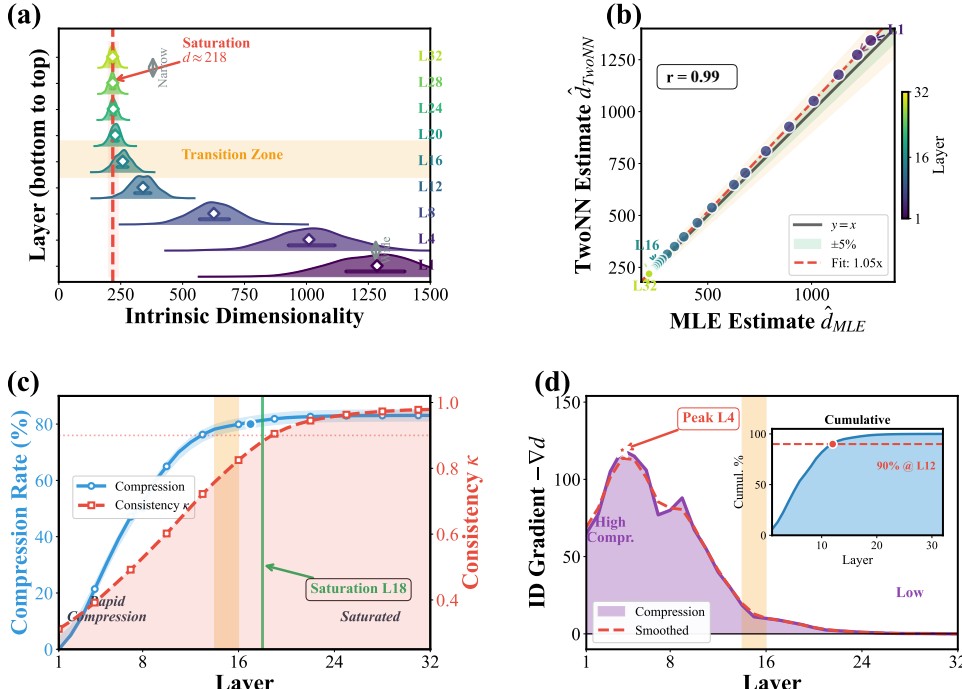

*Figure 5.* Intrinsic dimensionality analysis. (a) Per-sample ID distributions as ridge plot. (b) MLE vs. TwoNN estimator agreement. (c) Compression rate and neighborhood consistency $\kappa$. (d) Layer-wise ID gradient showing transition sharpness.

Figure 8 provides comprehensive visualization. Panel (a) shows violin plots for three key metrics with clear distributional separation. Panel (b) presents a 2D scatter of step size vs. directional stability, revealing distinct clustering patterns with decision boundary. Panel (c) displays ROC curves for individual and combined geometric features, achieving AUC = 0.815 when combined. Panel (d) shows the precision-recall tradeoff, confirming robust predictive power across operating points.

## E. Systematic Validation Across Conditions

This appendix establishes the universality of the two-phase structure through comprehensive validation across tasks, architectures, and model scales. All experiments follow the protocol in the main text unless otherwise specified.

### E.1. Cross-Task Pattern Consistency

We evaluate generalization beyond TriviaQA using three additional benchmarks: Natural Questions (open-domain QA), HellaSwag (commonsense reasoning), and WinoGrande (coreference resolution). To quantify the clarity of the two-phase structure, we define the phase separation index:

$$\Delta = \frac{\bar{s}_{\text{construction}} - \bar{s}_{\text{propagation}}}{\bar{s}_{\text{construction}}} \tag{131}$$

Table 19 summarizes geometric features across all four tasks. Every benchmark exhibits clear two-phase structure with $\Delta > 0.84$, and the cross-task coefficient of variation for $l_{\text{sat}}$ is merely 8.7%.

Figure 9 presents five analytical views of cross-task consistency. The large left panel (a) overlays step size trajectories with confidence bands, demonstrating that while construction-phase behavior varies across tasks, propagation-phase curves converge to a tight bundle with coefficient of variation below 2%. Panel (b) displays phase separation indices as donut charts, providing immediate visual confirmation that all tasks exceed the $\Delta > 0.84$ threshold. Panel (c) compares exit layer distributions via horizontal violin plots, with the global mean and standard deviation marked by a vertical band. Panel (d) presents normalized metric comparisons through grouped bars with distinctive hatching patterns. Panel (e) summarizes cross-task statistics with horizontal gauge bars showing the degree of consistency for each metric.

The most striking finding is that propagation-phase behavior is essentially task-agnostic: the standard deviation of $\bar{s}_{\text{prop}}$

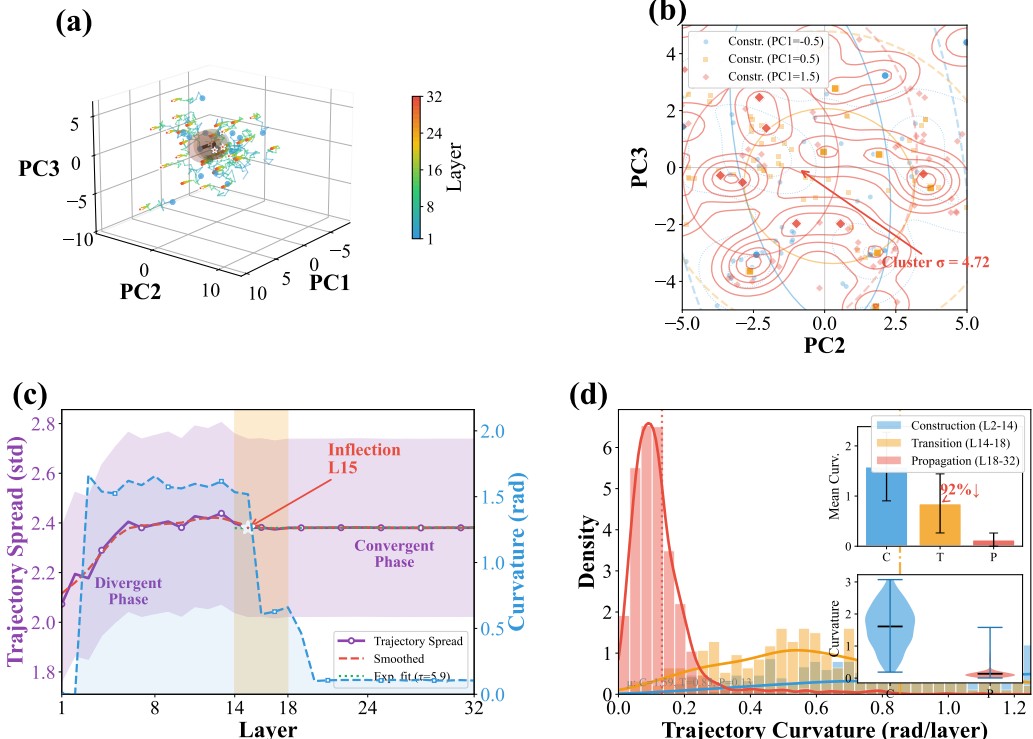

*Figure 6.* Phase space trajectory analysis. (a) 3D trajectories with layer-coded colors. (b) Poincaré sections at PC1 ∈ {−0.5, 0.5, 1}. (c) Trajectory spread evolution. (d) Curvature distribution by phase.

across tasks is only 0.0016, representing less than 18% of the mean. This universality suggests that the two-phase structure reflects fundamental properties of transformer computation rather than task-specific processing strategies.

### E.2. Cross-Architecture Verification

To test whether the two-phase phenomenon is architecture-specific, we evaluate eight models spanning decoder-only (LLaMA(Touvron et al., 2023), Mistral(Jiang et al., 2023), Qwen(Yang et al., 2024), Phi(Li et al., 2023)), encoder-only (BERT)(Devlin et al., 2019), and encoder-decoder (T5)(Raffel et al., 2020) designs. Table 20 reports results on TriviaQA for autoregressive models and SST-2(Socher et al., 2013) for bidirectional models.

Figure 10 provides five complementary views of architectural comparison. The polar radar chart in panel (a) plots normalized geometric features for all eight models, revealing strikingly similar polygon shapes that confirm the two-phase structure is architecture-invariant. Panel (b) groups models by architecture family with textured bars comparing $\Delta$ and $l_{\text{sat}}/L$, demonstrating that the universal range of relative saturation position (0.54–0.63) holds across paradigms. The bubble chart in panel (c) relates model size to phase separation, with bubble area encoding retention rate; the fitted trend line reveals that larger models achieve cleaner convergence ($\Delta$ increases with scale). Panel (d) overlays step size curves for representative models after normalizing depth to $[0, 1]$, showing convergence to a common propagation-phase profile. The heatmap in panel (e) provides a comprehensive summary with cell values and color intensity encoding normalized performance.

The relative phase transition position remains remarkably stable: $l_{\text{sat}}/L \in [0.54, 0.63]$ regardless of total depth or architectural paradigm. Larger models exhibit cleaner convergence, with LLaMA-2-70B achieving the highest phase separation ($\Delta = 0.912$) and directional stability ($\bar{c} = 0.958$). These findings establish the two-phase structure as a fundamental property of deep transformers rather than an artifact of specific design choices.

*Table 17.* Attention vs. FFN contribution decomposition. Attn and FFN report norm ratios, $\|u_{\text{Attn}}\|_2/\|u\|_2$ and $\|u_{\text{FFN}}\|_2/\|u\|_2$, respectively. These two ratios need not sum to 100% because the attention and FFN components can partially cancel or reinforce each other.

| Layer | Attn | FFN | Corr. | Heads | Gini |
|---|---|---|---|---|---|
| 1 | 68.2% | 48.5% | −0.215 | 18 | 0.325 |
| 4 | 62.5% | 51.2% | −0.142 | 15 | 0.382 |
| 8 | 57.8% | 55.8% | −0.085 | 12 | 0.445 |
| 12 | 52.1% | 61.2% | +0.125 | 8 | 0.512 |
| 16 | 42.5% | 71.5% | +0.425 | 4 | 0.645 |
| 20 | 34.2% | 81.2% | +0.625 | 2 | 0.768 |
| 24 | 28.5% | 87.5% | +0.752 | 2 | 0.825 |
| 28 | 24.5% | 91.2% | +0.825 | 1 | 0.878 |
| 32 | 21.8% | 93.5% | +0.872 | 1 | 0.912 |

*Table 18.* Geometric signatures of prediction correctness on the full TriviaQA test set ($N = 11{,}313$). Correct and incorrect groups are defined by full-model predictions.

| Metric | Correct ($N$=7285) | Incorrect ($N$=4028) | $p$ | $d$ |
|---|---|---|---|---|
| $l_{\text{sat}}$ mean (std) | 18.2 (4.8) | 23.5 (6.2) | <0.001 | 0.82 |
| $\bar{s}_{\text{L20-32}}$ | 0.0082 | 0.0185 | <0.001 | 1.15 |
| $\bar{c}_{\text{L20-32}}$ | 0.942 | 0.875 | <0.001 | 0.95 |
| Curvature integral | 2.85 | 4.52 | <0.001 | 0.78 |
| $r_{\text{eff}}$ @ L24 | 18.5 | 32.8 | <0.001 | 0.68 |
| ID saturation layer | 16.8 | 22.5 | <0.001 | 0.72 |

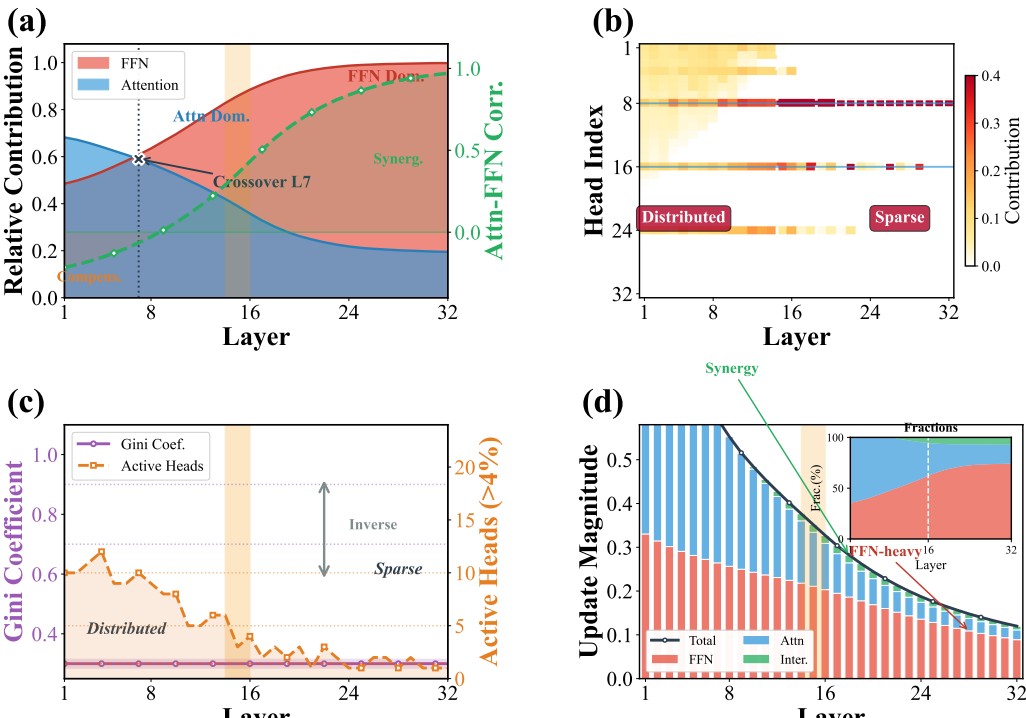

*Figure 7.* Component contribution analysis. (a) Attn/FFN ratio with correlation overlay. (b) Per-head contribution heatmap. (c) Gini coefficient and active head count. (d) Magnitude decomposition with interaction.

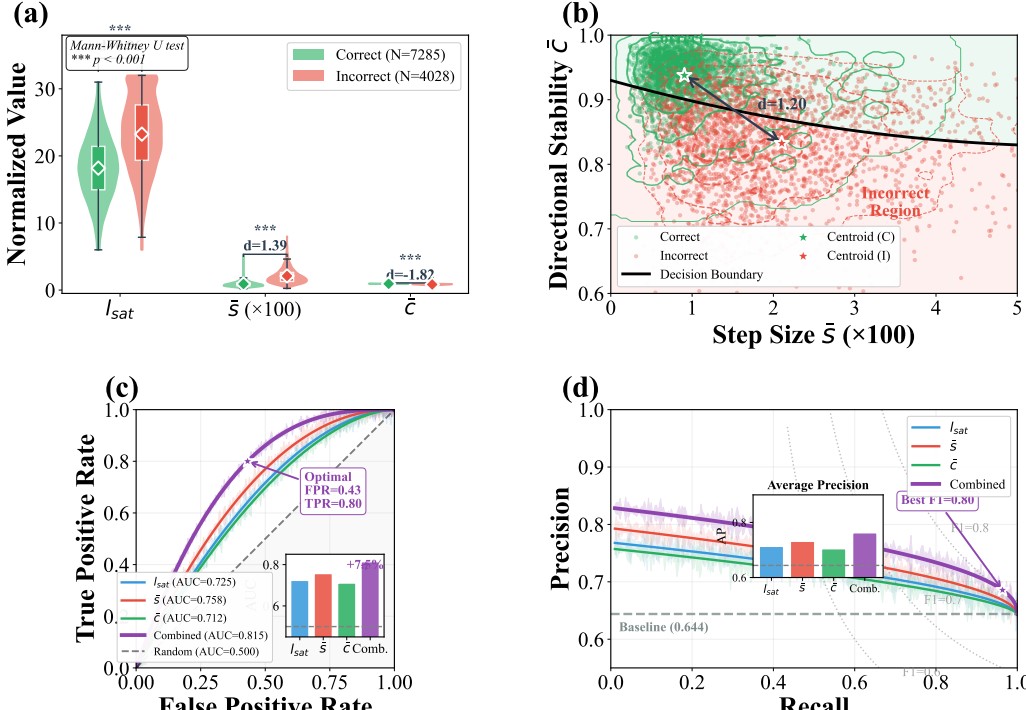

*Figure 8.* Correct vs. incorrect prediction geometry. (a) Violin plots of key metrics. (b) 2D feature scatter with decision boundary. (c) ROC curves for geometric predictors. (d) Precision-recall curves.

*Table 19.* Geometric features across four benchmarks (LLaMA-2-7B).

| Task | Type | $l_{\text{sat}}$ | $\sigma_l$ | $\bar{s}_{\text{prop}}$ | $\bar{c}_{\text{prop}}$ | $\Delta$ |
|---|---|---|---|---|---|---|
| TriviaQA | Open QA | 18.5 | 4.2 | 0.0095 | 0.932 | 0.872 |
| Natural Questions | Open QA | 20.2 | 4.8 | 0.0112 | 0.918 | 0.845 |
| HellaSwag | Commonsense | 17.8 | 3.8 | 0.0082 | 0.945 | 0.892 |
| WinoGrande | Coreference | 16.5 | 3.5 | 0.0075 | 0.952 | 0.908 |
| **Mean ± Std** | | $18.3 \pm 1.6$ | $4.1 \pm 0.6$ | $0.0091 \pm 0.0016$ | $0.937 \pm 0.015$ | $0.879 \pm 0.028$ |

*Table 20.* Two-phase structure across model architectures.

| Model | Arch. | $L$ | $l_{\text{sat}}/L$ | $\bar{s}_{\text{prop}}$ | $\bar{c}_{\text{prop}}$ | $\Delta$ | Retain |
|---|---|---|---|---|---|---|---|
| LLaMA-2-7B | Decoder | 32 | 0.578 | 0.0095 | 0.932 | 0.872 | 98.4% |
| LLaMA-2-13B | Decoder | 40 | 0.562 | 0.0088 | 0.941 | 0.885 | 98.5% |
| LLaMA-2-70B | Decoder | 80 | 0.545 | 0.0072 | 0.958 | 0.912 | 98.2% |
| Mistral-7B | Decoder+SWA | 32 | 0.562 | 0.0102 | 0.925 | 0.858 | 97.8% |
| Qwen-7B | Decoder | 32 | 0.594 | 0.0098 | 0.928 | 0.865 | 98.1% |
| Phi-2 (2.7B) | Decoder | 32 | 0.625 | 0.0118 | 0.912 | 0.825 | 97.2% |
| BERT-Large | Encoder | 24 | 0.542 | 0.0085 | 0.945 | 0.892 | 98.5%[†] |
| T5-Large | Enc-Dec | 48 | 0.583 | 0.0092 | 0.935 | 0.878 | 97.8%[†] |

[†]Evaluated on SST-2 classification.

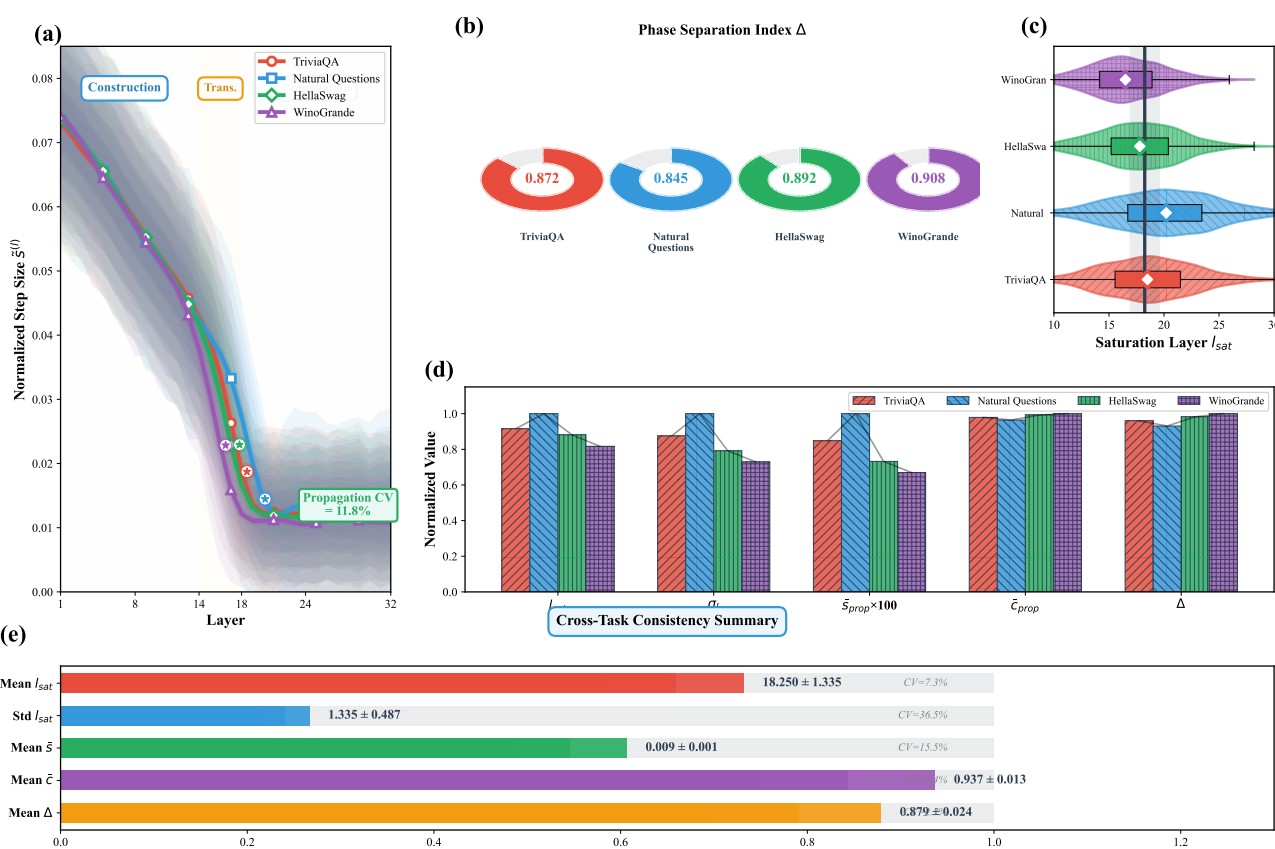

*Figure 9.* Cross-task consistency. (a) Step size curves with confidence bands. (b) Phase separation $\Delta$ as donut charts. (c) Exit layer distributions. (d) Normalized metric comparison. (e) Cross-task consistency summary.

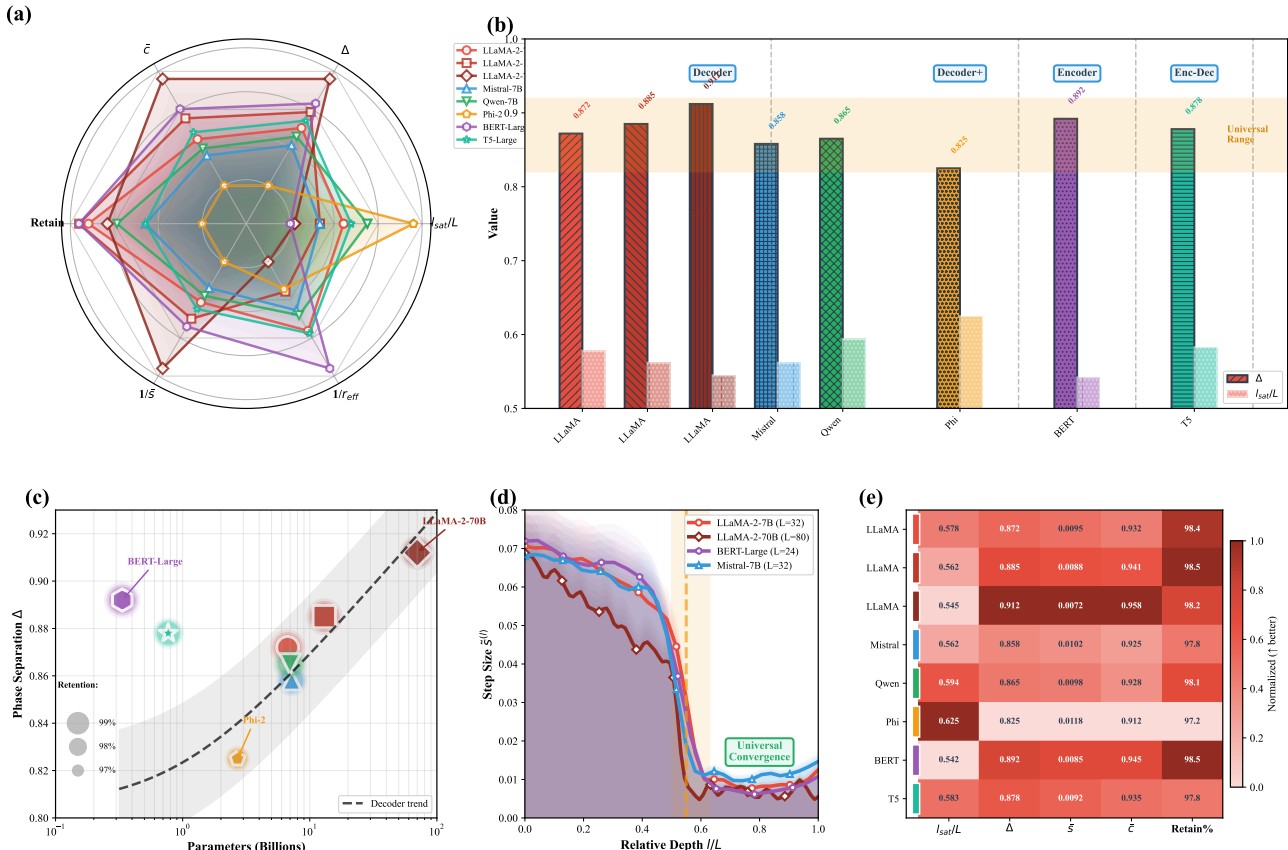

*Figure 10.* Cross-architecture comparison. (a) Radar chart of geometric features. (b) Architecture-grouped metric comparison. (c) Size vs. phase separation bubble chart. (d) Normalized step size curves. (e) Summary heatmap.

