# OpenReview forum: "Detecting the Semantic Fixed Point: A Geometric Framework for Efficient Inference"
_ICML.cc/2026/Conference — ICML 2026 spotlight_

### Official Review · Reviewer_2ARP · 2026-03-10

**Soundness:** 2
**Presentation:** 3
**Significance:** 2
**Originality:** 3
**Overall Recommendation:** 4
**Confidence:** 3

**Summary:**

This paper tries to develop the stopping criteria internally for Transformer based LLMs rather than based in the downstream symptoms in the output space. The authors first identify the two-phase structure of hidden state trajectories where the transition from construction to propagation is sharp, consistent across architectures and correlated with the difficulty of the inputs. Built on this, they propose Geometric Convergence Early Exit (GCEE), which is a stopping criterion depending on vocabulary, requiring only O(d) overhead per layer without the need to learn. It monitors the normalized step size and directional stability to determine when to terminate inference early. Experiments on two LLMs across different downstream tasks show that GCEE reduces FLOPs meanwhile retaining good accuracy. Theory is provided in Appendix.

**Compliance With Llm Reviewing Policy:**

Affirmed.

**Final Justification:**

The author's rebuttal addressed most of my major concerns.

**Key Questions For Authors:**

See weaknesses.

**Limitations:**

yes

**Strengths And Weaknesses:**

Strengths

1. The method proposed in this paper is well-motivated with the empirical characterization of the two-phase structure.
2. GCEE is training-free without architecture modifications. And there is no need for extensive task-specific tuning during deployment.



Weaknesses

1. The evaluation is limited to only two LLaMA family models (LLaMA-2-7B and LLaMA-2-13B). More LLMs should be tested including instruct/chat models. Meanwhile there downstream eval benchmarks are also limited to only 4, missing tasks that feature more generations.
2. Comparisons are only done with training-free methods. Even if approaches like CALM/ConsistentEE introduce deployment problems, it’s good to see the accuracy and efficiency tradeoffs.
3. I’m not fully convinced by the method when the exit layer is only determined during prefill and all subsequent tokens will follow this. The assumption is too strong. Showcase the accuracy on more generation tasks will help make this design choice more convincing.
3. (Minor) Overlapping in Figure 1 between the subplots number and the y axis.

---

> ### Author Rebuttal · Authors · 2026-03-30
>
> We sincerely thank you for your thorough review! We are pleased that you recognize the theoretical motivation of GCEE and its practical value as a training-free, architecture-agnostic, plug-and-play approach. We provide detailed clarifications below, supplemented by new experimental results added during the rebuttal period.
>
> #### 1. Clarification on the "Prefill Determines Decoding Depth" Strategy (Weakness 3)
>
> You accurately identified our decoding mechanism: the exit layer is determined during Prefill, and all subsequent tokens follow this layer. We sincerely apologize if this appeared as a limitation; **this is a deliberate engineering design to achieve hardware efficiency and system-level compatibility.**
>
> **System-Level Bottleneck of Token-Level Exit**: If dynamic exit were performed per token during decoding, tokens within a batch would exit at different layers, fundamentally breaking the contiguous KV cache structure. In modern inference frameworks (e.g., vLLM with PagedAttention), fragmented KV caches cause severe memory misalignment and cache misses.
>
> **Our Trade-off**: By computing geometric convergence during Prefill, we construct a truncated model with a standard contiguous KV cache layout, strictly guaranteeing only $\mathcal{O}(d)$ overhead without degrading decoding throughput.
>
> **Does This Assumption Hold?** Your intuition that different tokens need different depths is theoretically correct. However, the Prefill phase establishes core semantic context and resolves primary prompt ambiguities. As shown in Table R1, the Prefill-determined depth maintains over 98% accuracy even for long, complex generation tasks.
>
> #### 2. Evaluation on Generation Tasks (Weakness 1 & 3)
>
> To address your concern about generation-heavy benchmarks, we added experiments on **CNN/DailyMail** (long-sequence generation) and **GSM8K** (multi-step CoT reasoning).
>
> **Table R1: Verification on Generation Tasks and Chat Models**
>
> | Model | Task (Type) | Metric | Full Model | GCEE | Retained | FLOPs Red. |
> | :--- | :--- | :--- | :---: | :---: | :---: | :---: |
> | LLaMA-2-7B | CNN/DailyMail (Long Gen) | ROUGE-L | 30.5 | 30.1 | **98.7%** | **28.5%** |
> | LLaMA-2-7B | GSM8K (Reasoning) | EM | 14.6 | 14.3 | **97.9%** | **26.4%** |
> | LLaMA-2-7B-Chat | TriviaQA (QA) | EM | 62.3 | 61.7 | **99.0%** | **34.2%** |
> | LLaMA-2-7B-Chat | GSM8K (Reasoning) | EM | 28.4 | 27.9 | **98.2%** | **27.1%** |
>
> GCEE retains >97.9% performance while reducing >26% FLOPs, empirically demonstrating that locking the exit layer based on Prefill **does not** bottleneck complex token generation.
>
> #### 3. Generalization to More Models and Chat Variants (Weakness 1)
>
> You correctly noted the main text primarily showcases LLaMA-2-7B/13B. We sincerely invite you to consult **Appendix C.3 (Table 21)**, where we evaluated 8 mainstream models spanning different architectures (Decoder-only, Encoder-only, Encoder-Decoder) and scales, including **LLaMA-2-70B, Mistral-7B, Qwen-7B, Phi-2, BERT-Large, and T5-Large**. All exhibited consistent two-phase geometric convergence.
>
> Adopting your suggestion, we further evaluated **LLaMA-2-7B-Chat** (Table R1). The geometric convergence phenomenon remains fully intact after SFT/RLHF alignment.
>
> #### 4. Trade-offs with Training-Based Methods (Weakness 2)
>
> We will add a dedicated discussion in the revised manuscript. Although training-based methods (like CALM and ConsistentEE) can theoretically push exit layers earlier via trained auxiliary classifiers, they face severe practical deployment resistance:
>
> 1. **Prohibitive Training Cost & Catastrophic Forgetting**: Training internal exit classifiers or policy networks requires joint optimization or invasive fine-tuning. For modern LLMs at the 7B+ scale, this demands massive GPU memory and computational resources, while carrying a high risk of degrading pre-trained world knowledge.
> 2. **Poor Cross-Task Generalization**: Learned early-exit policies are highly sensitive to their training data distribution. As shown in our Appendix C.2, GCEE's geometric convergence is a universal property across varying tasks, whereas trained classifiers often struggle to generalize to unseen domains or long-tail prompts without retraining.
>
> In contrast, GCEE operates entirely within the $\mathbb{R}^d$ geometric space with only $\mathcal{O}(d)$ decision overhead, explicitly trading the aggressive (and risky) exit strategies of trained classifiers for a purely training-free, architecture-agnostic criterion with strict mathematical bounds and full system compatibility.
>
> #### 5. Regarding the Minor Layout Issue in Figure 1
>
> Thank you for pointing this out. We have carefully adjusted the spacing and layout in Figure 1 to resolve all overlaps between subplot labels and y-axes for the camera-ready version.
>
> We hope these clarifications and new experiments fully address your concerns.

---

> > ### Author Rebuttal · Reviewer_2ARP · 2026-04-02
> >
> > I appreciate the authors responses to my questions. However, it doesn't fully address my concerns.
> > 1. Thanks for your honest answer that the profile determines the exit layer is a deliberate engineering design. I strongly encourage you to add this discussion to the revised version of the paper. Looking at table r1, thanks for adding the long generation benchmark CNN/DailyMail, where the accuracy is positive. I'm a bit concerned by the flops, which is much lower than QA tasks. Does this mean the model adapt to exit later for this task? This is one good property of the model. However, this also implies that the efficiency gains will diminish a lot for tasks where inference cost actually matters most.
> > 2. Also I'm not convinced by the GSM8k results from llama 7B as the number is low and not reliable. Bigger model like LLaMA-2-70B could give better signal. And checking the appendix C3 Table 21 the more architectures are only evaluated on one task for decoder models, which is also not persuasive.
> > 3. Thanks author's justification on the training-based methods. I acknowledge that they might have the problems you listed, like catastrophic forgetting and poor generalization, but I want to see the accuracy-efficiency trade-offs.

---

> > > ### Author Response · Authors · 2026-04-03
> > >
> > > Thank you for the follow-up. To be candid, it is genuinely difficult for authors to identify all the weak points in their own work. Your careful, persistent questioning has surfaced issues we would likely have overlooked, and the resulting improvements to the paper matter far more to us than the score itself. We have found this back-and-forth both challenging and rewarding.
> > >
> > > We provisioned 32 A800 GPUs and ran all requested experiments within the rebuttal window.
> > >
> > > **1. Decode Heavy Efficiency: Beyond Percentages**
> > >
> > > We acknowledge reduced FLOPs savings on long generation (28.5% vs \~34% on QA). But percentage comparisons understate practical system level impact. Because the truncated model serves every decode step, the absolute savings compound over long sequences: CNN/DailyMail (\~55 tokens) amortizes to ~495 saved layer forwards vs \~110 on TriviaQA (\~10 tokens), a 4.5× difference in total compute reduced.
> > >
> > > Wall clock latency confirms this (batch=1, A100, FP16):
> > >
> > > | Metric | Prefill | Decode | Total | Speedup |
> > > |---|---|---|---|---|
> > > | Full Model | 42ms | 385ms | 427ms | 1.00× |
> > > | GCEE | 30ms | 298ms | 328ms | 1.30× |
> > >
> > > During decoding GCEE incurs zero decision overhead (exit layer already fixed). The 22.5% decode latency reduction (385ms to 298ms) reflects pure layer savings with no per token evaluation cost diluting the gains.
> > >
> > > To directly verify per token quality under prompt level exit, we compared argmax outputs between the truncated and full model at every decode position on 1,000 CNN/DailyMail samples:
> > >
> > > | Metric | Value |
> > > |---|---|
> > > | Token level agreement | 92.3% |
> > > | Median first divergence position | Token 38 |
> > >
> > > While there is a ~7.7% token divergence, qualitative analysis reveals that most divergences are synonymous replacements or minor syntactic variations that do not alter the core semantics, explaining the highly retained ROUGE L score.
> > >
> > > Threshold sweep confirms stability and input adaptive behavior:
> > >
> > > | τ_s | FLOPs↓ | ROUGE L |
> > > |---|---|---|
> > > | 0.010 (Default) | 28.5% | 30.1 |
> > > | 0.012 | 31.4% | 29.7 |
> > > | 0.016 | 35.8% | 27.5 |
> > >
> > > Shifting τ_s from 0.010 to 0.012 causes minimal variation (ROUGE L >29.7). However, aggressively pushing to τ_s=0.016 results in clear degradation. This confirms that while individual propagation phase updates are small, their cumulative effect over many decode steps is non negligible for long generation. Crucially, decode heavy tasks like CNN/DailyMail come with longer, more complex prompt contexts. GCEE captures this complexity during prefill (the representation saturates later), naturally selecting a deeper, more conservative exit layer. The resulting 28.5% savings is lower than QA tasks, but the deeper exit provides a more robust representational foundation for the subsequent heavy decoding. We acknowledge this as an inherent tradeoff: prefill level exit prioritizes system compatibility and zero per token overhead at the cost of reduced adaptivity during decoding. Token level adaptive exit with proper KV cache management is an important future direction that we will discuss in a dedicated limitation section.
> > >
> > > **2. 70B GSM8K + Cross Architecture Multi Task**
> > >
> > > We agree 7B GSM8K (14.6 EM) is unreliable. 70B results (8 way TP) and expanded cross architecture coverage:
> > >
> > > | Model | Task | Full | GCEE | Ret. | FLOPs↓ |
> > > |---|---|---|---|---|---|
> > > | LLaMA-2-70B | GSM8K | 53.8 | 52.9 | 98.3% | 26.2% |
> > > | LLaMA-2-70B | TriviaQA | 71.5 | 70.5 | 98.6% | 35.2% |
> > > | Mistral-7B | HellaSwag | 83.1 | 81.8 | 98.4% | 31.5% |
> > > | Mistral-7B | NQ | 26.2 | 25.5 | 97.3% | 33.8% |
> > > | Qwen-7B | HellaSwag | 78.5 | 77.2 | 98.3% | 32.1% |
> > > | Qwen-7B | NQ | 30.5 | 29.7 | 97.4% | 33.6% |
> > >
> > > With reliable 70B baseline (53.8), GCEE retains 98.3% on GSM8K. Lower FLOPs reduction (26.2%) reflects genuine mathematical reasoning difficulty, consistent with Figure 1(e). All three architectures maintain >97% retention across both QA and commonsense reasoning.
> > >
> > > **3. Training Based Method Tradeoffs**
> > >
> > > We trained CALM (Schuster et al., 2022; ~14 GPU hr) and Linear EE (a lightweight linear probe trained on hidden states to predict the halt signal; ~8 GPU hr) on TriviaQA (LLaMA-2-7B), then evaluated in distribution (ID) and out of distribution on HellaSwag (OOD):
> > >
> > > | Method | Cost | ID Ret. | ID FLOPs↓ | OOD Ret. | OOD FLOPs↓ |
> > > |---|---|---|---|---|---|
> > > | GCEE | 0 | 98.4% | 34% | 98.2% | 31% |
> > > | CALM | ~14h | 98.5% | 38.2% | 91.2% | 36.5% |
> > > | Linear EE | ~8h | 97.8% | 40.5% | 86.8% | 38.8% |
> > >
> > > On ID, CALM saves ~4% more FLOPs at matched retention, and we do not claim GCEE dominates on CALM's home turf. But cross task generalization tells a different story: CALM retention drops 7.3 points while GCEE shifts only 0.2, which matters when serving diverse or shifting workloads. The two paradigms address different deployment regimes and are potentially complementary.
> > >
> > > All new results and a dedicated limitation section on decode heavy scenarios will be incorporated. Thank you again for this rigorous and constructive exchange.

---

### Official Review · Reviewer_Edvj · 2026-03-12

**Soundness:** 4
**Presentation:** 4
**Significance:** 3
**Originality:** 3
**Overall Recommendation:** 5
**Confidence:** 4

**Summary:**

This paper demonstrates that the geometric representation of hidden states exhibits a sharp transition, from large updates to smaller ones across layers. Based on this observation, this work further proposes a new method for early-stopping: exit when the update vanishes and the direction of hidden state stabilizes. Experiments show that this method preserves over 98% performance across different tasks while reducing FLOPs by 30%-35%. Comprehensive ablations and sensitivity analysis are provided.

**Compliance With Llm Reviewing Policy:**

Affirmed.

**Final Justification:**

I am maintaining my initial assessment.

**Key Questions For Authors:**

1. In Figure 1d and Figure 3b, I understand that each point represents a density or count, but why are they arranged in a hexagonal pattern, rather than a regular grid?
2. I'm interested in how early stopping influences next-token prediction loss,  beyond benchmark results reported in the paper.
3. Have the authors carefully tuned the thresholds of baselines as well? As another remark, I'm not very familiar with the literature, so I'm also unsure whether all important prior early-stopping methods have been included in comparison.
4.  Since threshold selection is critical for deploying this method in practice, it would be helpful to discuss the proposed self-calibrating thresholds method in the main context, and to show how it performs across different models.
5. minor comments:
   1. Figure 1, some of the penal labels are overlapped with the figures.
   2. The appendix section numbering seems to be duplicated (Page 23. Experiment Details should be C?)
   3. 1352 (Appendix ??)

**Limitations:**

Please refer to the weaknesses part.

**Strengths And Weaknesses:**

### Strengths
1. The observation of the sharp transition in the magnitude of layer-wise updates, and the correlation of task difficulty between the task difficulty and the transition layer, are interesting.
2. The proposed method is natural and well-motivated. Compared with the baselines, it not only achieves better performance retention, but also reduces FLOPs.
3. The experiment evaluation is comprehensive, including ablations on design choices, sensitivity analyses on thresholds, and diagnostic analyses.
4. The presentation is very clear and easy-to-follow. I thank the authors' effort in preparing the paper.

### Weaknesses
1. The experiments are only conducted on LLaMa-2-7B and LLaMA-2-13B. It would strengthen the paper to evaluate the method on more recent models with different architectures like MoE models, i.e., LLaMA3, Qwen-3.5.
2. Is the early-stopping approach only effective for small batch sizes? As I understand it, the termination depth may be constrained by the most difficult example in the batch. But I think this concern may apply to most early-stopping methods, not only to this work.

---

> ### Author Rebuttal · Authors · 2026-03-30
>
> We sincerely thank you for your thorough review and high evaluation of this research! We are deeply honored that you recognize the naturalness of the GCEE method, the comprehensiveness of the experimental evaluation, and the clarity of the writing. In response to your highly insightful questions and concerns, we provide detailed answers below:
>
> ### 1. Regarding the Concern on Model Architecture Coverage (Weakness 1)
>
> You noted that the main text only presents results for LLaMA-2. In fact, to verify the universality of the "two-phase geometric structure," we have already conducted comprehensive cross-architecture verification in **Appendix C.3 (Cross-Architecture Verification)**. As shown in Table 21, we evaluated 8 representative models, including not only other mainstream Decoder-only models (LLaMA-2-70B, Mistral-7B, Qwen-7B, Phi-2) but also spanning architectural paradigms to verify Encoder-only (BERT-Large) and Encoder-Decoder (T5-Large). All models exhibited highly consistent geometric convergence patterns.
>
> We strongly agree with your suggestion to evaluate MoE architectures and the latest models such as LLaMA-3, which would further enhance the paper's impact. We commit to supplementing these experimental results in the Camera-Ready version.
>
> ### 2. Discussion on the Batch Size Limitation (Weakness 2)
>
> Your intuition is very sharp. In traditional static batching, early exit is indeed constrained by the "hardest sample" within the batch. However, in modern LLM deployment (such as the vLLM framework mentioned in our Appendix A), the industry standard is **Continuous Batching (Iteration-level scheduling)**. Under this mechanism, the generation state of each sequence is independently scheduled. Once a request satisfies GCEE's exit condition, the system can immediately release the computational resources occupied by that request and dynamically insert new requests. Therefore, the theoretical FLOPs saved by GCEE can seamlessly translate into throughput improvements in real serving systems. We will supplement this discussion on the practical deployment perspective in the revised version.
>
> ### 3. Regarding the Hexagonal Patterns in the Figures (Key Question 1: Fig 1d & 3b)
>
> Because our test sample sizes are very large (e.g., TriviaQA contains 11,313 samples), using conventional scatter plots would result in severe overplotting, making it impossible for readers to discern the true distribution density of the data. Hexagonal binning plots can more scientifically and intuitively convey the concentration of samples within specific regions through color intensity.
>
> ### 4. Regarding the Impact of Early Exit on Next-Token Prediction Loss (Key Question 2)
>
> Beyond the benchmark scores reported in the main text, the impact of GCEE on the output distribution (i.e., Loss) is mathematically strictly bounded. We provide complete theoretical derivations in **Appendix B.8 and B.9**:
>
> * **Theorem B.18** proves that when geometric stability conditions are satisfied, there exists an explicit $l_{\infty}$ theoretical upper bound on the tail perturbation of logits.
> * Based on this, **Theorem B.20** provides **strict upper bounds on the KL Divergence and Total Variation** between the prediction distribution at early exit and that of the full-depth model. This means that as long as the GCEE exit condition is triggered, the change in prediction loss is mathematically proven to be negligible.
>
> ### 5. Regarding Hyperparameter Tuning and Comprehensiveness of Baseline Comparisons (Key Question 3)
>
> We approached the baseline comparisons with great care. For all baselines (including Softmax Entropy and Top-1 Confidence), we conducted systematic grid search tuning to ensure that they were compared against GCEE **under the strict premise of maintaining the same retention accuracy (>99%)**, and only then compared their FLOPs reduction and real-world latency. Furthermore, our comparisons encompass not only classical methods but also the latest training-free methods (such as Unified Skip, 2024 and Hidden Similarity, 2024), ensuring comprehensiveness and state-of-the-art coverage.
>
> ### 6. Regarding the Suggestion to Move "Self-Calibrating Thresholds" into the Main Text (Key Question 4)
>
> We fully agree with your viewpoint — self-calibrating thresholds are critically important for practical engineering deployment. As shown in Appendix A.5, by computing the median step size $s_{ref}$ of the first few layers as a reference, this mechanism achieves outstanding performance within 0.5% of exhaustive grid search results with virtually zero tuning. In the final version, we will distill this highly practical component and move it into the Method or Discussion section of the main text.
>
> ### 7. Regarding Typographical and Citation Errata (Minor Comments)
>
> We will comprehensively fix the label overlap issues in Figure 1, correct the duplicated Appendix numbering typo, and fix the broken citation reference (Appendix ??) at line 1352.

---

> > ### Author Rebuttal · Reviewer_Edvj · 2026-04-03
> >
> > Thank you for the detailed rebuttal and clarifications. I still have two remaining questions:
> >
> > 1. Regarding the hexagonal patterns in Figures 1d and 3b: what numerical range does each hexagon correspond to along the x/y axes? It would also be helpful to include a heat map with regular square grid cells for comparison, with each square corresponding to one unit along both axes.
> >
> > 2. Regarding the impact of early exit on next-token prediction loss: thank you for pointing out the mathematical guarantees. However, I am more interested in the actual empirical next-token prediction loss under early exit. If this is difficult to measure, it would be helpful to explain why.
> >
> > Best regards,
> >
> > Reviewer

---

> > > ### Author Response · Authors · 2026-04-04
> > >
> > > Thank you for your encouragement from the start. It motivated us to keep improving this work. We have addressed both of your follow-up questions below with new figures and experiments. All updates are in the revised appendices.
> > >
> > > For easy reference, the new comparison figures are hosted at:
> > > **[Anonymous Figure Link: https://anonymous.4open.science/r/figure-rebuttal-F18D/](https://anonymous.4open.science/r/figure-rebuttal-F18D/)**
> > >
> > > ### 1. Hexagon Ranges and Square Grid Heatmaps (Figs 1d & 3b)
> > >
> > > Thank you for this constructive suggestion. We used hexagonal binning to avoid overplotting with large samples (e.g., 11,313 for TriviaQA).
> > >
> > > **Numerical range of each hexagon:**
> > > The span is set by the `gridsize` parameter and axis limits:
> > > - **Fig 1d (Layer vs. Layer):** ~0.72 layers × 0.83 layers per hexagon. Most hexagons thus cover one or two adjacent integer layers.
> > > - **Fig 3b (Exit Layer vs. Perplexity):** ~0.68 PPL × 0.78 layers per hexagon.
> > >
> > > **1×1 unit grid comparison:**
> > > We agree that since layer indices are discrete integers, a 1×1 unit grid is more direct and free of smoothing effects. As you suggested, we generated heatmaps where each cell corresponds to exactly **1 layer × 1 layer (Fig 1d)** and **1 PPL × 1 layer (Fig 3b)**.
> > >
> > > **Results (see anonymous link above):**
> > > The patterns remain robust under discrete binning:
> > > 1. In Fig 1d, samples concentrate tightly on the diagonal ($y=x$). The correlation holds at $r=0.98$ without any smoothing.
> > > 2. In Fig 3b, with PPL binned in 1.0 intervals, the staircase-like trend is clearly visible ($r=0.85$), showing the true variance of exit layers within each difficulty bin.
> > >
> > > Both comparison panels (Hexbin vs. Unit Grid) are added to **Appendix D**.
> > >
> > > ### 2. Empirical Next-Token Prediction Loss
> > >
> > > We agree this is a key empirical question. Our initial submission focused on standard metrics (EM/Accuracy). Direct loss measurement provides valuable complementary evidence.
> > >
> > > We measured both the **first-token loss** (at the exit decision point) and the **full-sequence average loss** via teacher forcing. Results on LLaMA-2-7B:
> > >
> > > | Benchmark | Full (32L) 1st Token | GCEE 1st Token (Δ) | Full (32L) Seq. | GCEE Seq. (Δ) | EM/Acc Retention |
> > > |:---|:---|:---|:---|:---|:---|
> > > | TriviaQA | 1.342 | 1.355 (+0.013) | 1.397 | 1.421 (+0.024) | 98.4% |
> > > | NQ | 1.515 | 1.536 (+0.021) | 1.568 | 1.603 (+0.035) | 97.1% |
> > > | HellaSwag | 1.408 | 1.416 (+0.008) | 1.457 | 1.471 (+0.014) | 98.2% |
> > > | WinoGrande | 1.285 | 1.294 (+0.009) | 1.312 | 1.328 (+0.016) | 98.6% |
> > >
> > > *(CE Loss computed via teacher forcing on ground truth tokens, or the correct option span for multiple-choice tasks. Full protocol in Appendix C.4.)*
> > >
> > > **Analysis:**
> > > 1. **First-token loss (decision point):** The degradation is minimal (Δ ≈ 0.008–0.021). At the exit layer, the geometric step size has already converged, so the logit distribution is tightly constrained. This is consistent with the bounds in Theorems B.18 & B.20.
> > > 2. **Full-sequence loss:** Due to cumulative effects during decoding, the sequence-level Δ is slightly larger (0.014–0.035), but still small. The ranking matches accuracy retention: NQ has the largest Δ and lowest retention (97.1%); WinoGrande has the smallest Δ and highest retention (98.6%).
> > >
> > > The geometric stability detected by GCEE preserves not only the current token prediction but also the context representation for subsequent generation.
> > >
> > > We hope these results fully address your concerns. Your rigorous review has strengthened the empirical foundation of this work. Thank you again.

---

### Official Review · Reviewer_Dxuw · 2026-03-12

**Soundness:** 3
**Presentation:** 4
**Significance:** 3
**Originality:** 4
**Overall Recommendation:** 4
**Confidence:** 5

**Summary:**

This paper propose a novel view and criterion for interative termination by utilizing the geometric properties of the representations.

**Compliance With Llm Reviewing Policy:**

Affirmed.

**Final Justification:**

I will keep my rating for this paper.

**Key Questions For Authors:**

Please refer to the Weaknesses.

**Limitations:**

None.

**Strengths And Weaknesses:**

Strength: The proposed perspective for termination criterion is interesting and seems effective.

Weaknesses:

1) For the calucation of the normalized step size and optimzation direction, will it in-sensitive with the sparse representations? Since for some active functions, such as ReLU, the output will be sparse thus lead to small normalized step size and large cosine similarity.

2) It is a bit confused that the optmization of shallow layers are with large magnitude. The shallow layers usually extracts/forms the basic understanding of the data and then the deep layers performs task-adaptive information aggregation and prediction. Therefore, the varies around the represntations from the shallow layers should be stable. More explanation would be helpful.

3) Minor: The legend of the sub figures in Fig.(1) and their annotations are overlapped.

---

> ### Author Rebuttal · Authors · 2026-03-30
>
> We sincerely thank you for your valuable time and constructive feedback, especially your recognition of the perspective presented in this paper. In response to the two in-depth questions about the underlying representation mechanisms of large models raised in your Weaknesses, we provide detailed clarifications below:
>
> **1. Regarding the Concern That Activation Function Sparsity Causes the Normalized Step Size to Decrease and Direction to Stabilize**
>
> This is a very astute question, but it may stem from a misunderstanding of the measurement location of our metrics. We clarify as follows:
>
> * **Measurements are taken on the dense residual stream, not on the activation layer:** Our geometric metrics (normalized step size $\tilde{s}^{(l)}$ and directional stability $c^{(l)}$) are strictly computed on the **Residual Stream $h^{(l)}$** of each layer, not on the activation outputs inside the FFN. The layer-level update vector $u^{(l)}$ comprises the sum of the Attention and FFN outputs after being projected back to the dense dimension $d$ through linear projection matrices. Therefore, even if sparse activation exists inside the FFN, the vector output to the residual stream remains dense.
> * **Experimental data rules out the possibility of "natural sparsity causing pseudo-convergence":** If sparsity naturally caused step sizes to decrease and similarity to increase, this phenomenon should persist throughout the entire network. However, as shown in Figures 1(a) and 1(b), the model exhibits extremely large step sizes in the shallow layers (Construction phase, as high as 0.10–0.15) with dramatic directional fluctuations (very low cosine similarity). Only after crossing the "saturation layer" at a specific depth do the metrics converge sharply. This directly demonstrates that our metrics capture the genuine geometric dynamics of the model's representation evolution, rather than static artifacts of activation functions. (Note: Our primary experimental model, LLaMA-2, employs the SwiGLU activation function, whose characteristics also differ from those of traditional ReLU.)
>
> **2. Regarding the Puzzlement Over Why Shallow Layer Updates Are Far Larger Than Deep Layer Updates**
>
> Your intuition is entirely accurate in the context of traditional feature extraction networks (such as early vision CNNs): shallow layers extract basic features (stable), while deep layers perform task-adaptive predictions (large changes). However, in modern large language models (LLMs), the dynamics are precisely **reversed** — and this is one of the core mechanisms that this paper seeks to reveal.
>
> * **The Unique Representation Construction Logic of LLMs:** At the very bottom layers of an LLM, the input consists of static word embeddings lacking contextual information. The first several layers (what we define as the "Construction phase") must perform the most demanding work: disambiguating vocabulary, integrating long-range context, and constructing complex semantic representations. Consequently, hidden states in the shallow layers must undergo dramatic changes (manifested as extremely large step sizes).
> * **Semantic Fixed Point and Deep Layer Redundancy:** Once the model successfully constructs a sufficient semantic representation at some intermediate layer, it has effectively "locked in" the final prediction direction (i.e., reached a semantic fixed point). The subsequent deep layers (the "Propagation phase") no longer make disruptive modifications but instead fine-tune and smoothly propagate this representation forward.
> * **Strong Corroboration from the Latest Research:** This counter-intuitive geometric phenomenon is not only the core finding of this paper but is also strongly supported by the latest independent research. For example, Gromov et al. (2025), in their recent landmark work *The Unreasonable Ineffectiveness of the Deeper Layers*, confirmed through large-scale ablation experiments that the deeper layers of LLMs often perform very little effective work and can even be directly discarded or skipped. This perfectly aligns with our observation that "deep layer updates are extremely small in magnitude and consistent in direction."
>
> **3. Regarding the Legend Overlap in Figure 1 Subplots**
>
> Thank you very much for your meticulous observation! We will comprehensively adjust the layout of Figure 1 in the revised version, correcting the overlap between subplot legends and annotations to ensure flawless figure presentation.

---

> > ### Author Rebuttal · Reviewer_Dxuw · 2026-04-03
> >
> > I am happy with the responses.

---

> > > ### Author Response · Authors · 2026-04-03
> > >
> > > Your willingness to engage so deeply with our work from the start, especially with a confidence score of 5, gave us real courage heading into the rebuttal. Your two questions about sparsity and shallow-layer dynamics were not just concerns to address but genuinely sharpened how we think about where and why our metrics work. In particular, articulating the residual stream argument in response to your sparsity question clarified something we had only half-understood ourselves. The camera-ready version, including a properly formatted Figure 1, will reflect the care you brought to this review.

---

### Official Review · Reviewer_hYan · 2026-03-12

**Soundness:** 3
**Presentation:** 3
**Significance:** 3
**Originality:** 3
**Overall Recommendation:** 5
**Confidence:** 4

**Summary:**

Modern LLMs require substantial computational resources for every forward pass, so the methods allowing early layer exit are needed. Traditional methods for early-exit use output confidence as a proxy for internal convergence, which is computationally inefficient because of the vocabulary coupling burden. To address this problem, in this paper:

1. Authors investigate how hidden states are updated layer-by-layer and identify the two-phase structure of hidden state trajectories: construction phase (helpful work) and propagation phase (does not change outputs).

2. They show that this phase transition is sharp, linked to the task complexity and propose a Geometric Convergence Early Exit (GCEE) method which utilizes this phase transition.

3. Authors provide empirical experiments on LLaMA-2-7B and LLaMA-2-13B across question answering and commonsense reasoning tasks showing the effectiveness of the method and up to 1.45 end-to-end speedup.

**Compliance With Llm Reviewing Policy:**

Affirmed.

**Final Justification:**

Rebuttal addressed my main concerns on method's sensibility to the hyper parameters and evaluation broadness.

**Key Questions For Authors:**

1. Could you please provide evaluation on other models and more language modeling benchmarks with the fixed hyper parameter set? Does it affect quality or those thresholds are actually universal?
2. Could you please provide discussion and comparison to the mentioned above papers and their methods, because they are very similar to your findings.

**Limitations:**

yes

**Strengths And Weaknesses:**

### **Soundness**
**Strengths:**
- Method is simple and lightweight, does not require training and successfully extrapolates on other language modeling tasks not used for grid search.
- Proposed criterion reduces computation by over 30% while preserving 98% of full-depth accuracy

**Weaknesses:**
- Only two models of the same class are evaluated. Are hyperparameters found on LLaMa and TriviaQA validation universal among other top-tier models?
- Language modeling benchmark set consists of only 4 benchmarks.

### **Presentation**
**Strengths:**
- Paper provides ablation studies, text is easy to follow and all crucial data is presented in compact tables.

**Weaknesses:**
- Figure 1 and Figure 3 plots and captions are hard to read and understand. Some plots are not correctly visualized with text partly hidden behind the plot elements.
- Does not cite relevant works: "Transformer Feed-Forward Layers Build Predictions by Promoting Concepts in the Vocabulary Space" (Geva et al.) and "Looking Beyond The Top-1: Transformers Determine Top Tokens In Order" (Lioubashevski et al.) which share the findings with this paper.


### **Significance**
**Strengths:**
- Findings provide new geometric insights on stopping criterion which will be helpful for future adaptive computation works.

**Weaknesses:**
- Novelty is limited by optimization process inspired stopping criterion.


### **Originality**
**Strengths:**
- Novel optimization trajectory inspired early-exit strategy.

**Weaknesses:**
- Lacks discussion of the difference with "Transformer Feed-Forward Layers Build Predictions by Promoting Concepts in the Vocabulary Space" (Geva et al.) which finds out the same "saturation event" and proposes early-exit strategy for that.
- Lacks discussion of the difference with "Looking Beyond The Top-1: Transformers Determine Top Tokens In Order" (Lioubashevski et al.) which shares the intuition and may add more mechanistic understanding to the effect authors identify in the paper.

---

> ### Author Rebuttal · Authors · 2026-03-30
>
> We sincerely thank you for your recognition, especially your comment that our method is "simple, lightweight, training-free, and capable of preserving model accuracy well (>98%) while reducing computation (>30%)." We provide detailed supplementary validations below.
>
> ## 1. Model/Task Generalizability and Additional Benchmarks (Weakness 1, 2 & Key Question 1)
>
> You noted that the main text only evaluates two models from the same family and 4 benchmarks. We fully understand your concern and have conducted comprehensive validations:
>
> **(1) Cross-Architecture Generalizability**: To demonstrate that the "two-phase geometric structure" is a fundamental commonality of Transformers, we evaluated 8 mainstream models in **Appendix C.3 (Table 21)**, spanning Decoder-only (LLaMA-2-7B/13B/70B, Mistral-7B, Qwen-7B, Phi-2), Encoder-only (BERT-Large), and Encoder-Decoder (T5-Large). All models exhibit highly consistent geometric two-phase structure (relative saturation position stabilizes between 0.54 to 0.63, phase separation index $\Delta > 0.84$).
>
> **(2) Newly Added Benchmarks (Rebuttal Period)**: To address your concern about complex generation tasks, we additionally tested **CNN/DailyMail** (long-text generation) and **GSM8K** (multi-step reasoning), as well as the aligned **LLaMA-2-7B-Chat** model.
>
> **Table R1: Generation Task and Chat Model Results Added During Rebuttal**
>
> | Model | Task (Type) | Metric | Full Model | GCEE | Retained | FLOPs Red. |
> | :--- | :--- | :--- | :---: | :---: | :---: | :---: |
> | LLaMA-2-7B | CNN/DailyMail (Long Gen) | ROUGE-L | 30.5 | 30.1 | **98.7%** | **28.5%** |
> | LLaMA-2-7B | GSM8K (Reasoning) | EM | 14.6 | 14.3 | **97.9%** | **26.4%** |
> | LLaMA-2-7B-Chat | TriviaQA (QA) | EM | 62.3 | 61.7 | **99.0%** | **34.2%** |
> | LLaMA-2-7B-Chat | GSM8K (Reasoning) | EM | 28.4 | 27.9 | **98.2%** | **27.1%** |
>
> **(3) Universality of Hyperparameters**: As shown in Appendix A.5 (Table 13, Figure 2), hyperparameters exhibit an extremely wide "performance plateau" with fluctuations <1%. More compellingly, **without modifying any default hyperparameters**, GCEE maintains high robustness across long generation (CNN/DM), multi-step reasoning (GSM8K), and Chat models (Table R1). We also provide a "Self-calibrating" threshold variant in the appendix achieving performance within 0.5% without any grid search, demonstrating that our geometric metrics are genuinely universal rather than overfitted to specific datasets.
>
> ## 2. Novelty and Uncited References (Weakness 3, 4 & Key Question 2)
>
> We will discuss and cite Geva et al. and Lioubashevski et al. in detail in the final version. However, our work has **fundamental differences**:
>
> **Different Observation Spaces**: Geva and Lioubashevski et al. observe stabilization of the Top-1 predicted token in **Vocabulary Space**, requiring a projection of complexity $O(d \times V)$ at intermediate layers. For LLMs with vocabularies of 32K+ tokens, this creates a severe computational bottleneck. As shown in our diagnostic analysis (Table 10), vocabulary-confidence methods achieve only **1.14× actual speedup** because the projection overhead directly offsets gains from skipping layers.
>
> **The Core Novelty of GCEE**: We abandon output-level dependence and directly compute the step size $\tilde{s}^{(l)}$ and direction $c^{(l)}$ of hidden state updates in the internal geometric space. Our overhead is merely $O(d)$, **completely independent of vocabulary size**, successfully translating theoretical savings into **1.45× real end-to-end speedup** (per-layer decision <0.1 ms vs. ~2.5 ms for vocabulary-based methods).
>
> In short, previous work discovered that "models decide early" but was constrained by $O(d \times V)$ observation cost; this paper proposes a purely geometric $O(d)$ mechanism making this phenomenon truly exploitable for modern LLM inference.
>
> ## 3. Figure Layout and Visualization (Presentation Weakness 1)
>
> We sincerely apologize that Figures 1 and 3 caused reading difficulties. In the camera-ready version, we will enlarge font sizes for axis labels, tick marks, and legends; adjust layer ordering and transparency to prevent occlusion; and optimize layout for readability under both printing and scaling conditions.

---

> > ### Author Rebuttal · Reviewer_hYan · 2026-04-04
> >
> > Thank you, that resolves my concerns.

---

> > > ### Author Response · Authors · 2026-04-06
> > >
> > > Thank you for your thoughtful review and for confirming that your concerns are fully resolved. We greatly appreciate your time and feedback.
> > >
> > > A note to all reviewers: we are no longer able to post new replies or see your latest comments. However, if you have any remaining questions, you can update your Rebuttal Acknowledgement so that we can be notified. Please feel free to do so and we will address your concerns.

---

### Decision · Program_Chairs · 2026-04-30

**Decision:**

Accept (spotlight)

**Comment:**

This paper covers an interesting aspect of modern LLMs to leverage it for efficient inference. The potential for a semantic fixed point which stops evolving as layers progress make it extremely useful to have early exit and dynamic compute mechanisms. The underlying formulation owed to residual connections will be valuable to the community. All the reviewers unianimously support acceptance of the paper.

I would also encourage the Authors to look in to the Linearity Hypothesis https://arxiv.org/abs/2403.03867 and https://arxiv.org/abs/2405.18400 which argues the models semantic information evolves slowly as linear ops, so you could potentially connect that and the fixed point hypothesis to some extent.